

# Towards universal unfolding of detector effects in high-energy physics using denoising diffusion probabilistic models

Camila Pazos[1]⋆, Shuchin Aeron[2,3]†, Pierre-Hugues Beauchemin[1,3]‡, Vincent Croft[4]◦, Zhengyan Huan[2,3]§, Martin Klassen[1]¶ and Taritree Wongjirad[1,3]∥

**1** Department of Physics and Astronomy, Tufts University, Medford, Massachusetts
**2** Department of Electrical and Computer Engineering,
Tufts University, Medford, Massachusetts
**3** The NSF AI Institute for Artificial Intelligence and Fundamental Interactions
**4** Leiden Institute for Advanced Computer Science LIACS, Leiden University, The Netherlands

⋆ camila.pazos@tufts.edu , † shuchin@eecs.tufts.edu , ‡ hugo.beauchemin@tufts.edu ,
◦ vincent.croft@cern.ch , § zhengyan.huan@tufts.edu ,
¶ martin.klassen@tufts.edu , ∥ taritree.wongjirad@tufts.edu

## Abstract

Correcting for detector effects in experimental data, particularly through unfolding, is critical for enabling precision measurements in high-energy physics. However, traditional unfolding methods face challenges in scalability, flexibility, and dependence on simulations. We introduce a novel approach to multidimensional object-wise unfolding using conditional Denoising Diffusion Probabilistic Models (cDDPM). Our method utilizes the cDDPM for a non-iterative, flexible posterior sampling approach, incorporating distribution moments as conditioning information, which exhibits a strong inductive bias that allows it to generalize to unseen physics processes without explicitly assuming the underlying distribution. Our results highlight the potential of this method as a step towards a "universal" unfolding tool that reduces dependence on truth-level assumptions, while enabling the unfolding of a wide range of measured distributions with improved adaptability and accuracy.

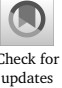

# 1  Introduction

Experimental data in high-energy physics (HEP) presents a distorted picture of the underlying physics processes due to detector effects. Unfolding is an inverse problem solved through statistical techniques that aims to correct the detector distortions of the observed data to make inferences about the true underlying distribution of particle properties. This process is essential for the validation of theories, constraining new physics models using experimental data, precision measurements, and comparison of experimental results between different experiments.

A standard approach to unfolding [1] begins with a predicted particle distribution $f_{\text{true}}(\mathbf{x})$, where $\mathbf{x}$ represents the true particle-level kinematic properties, that characterizes the underlying physics process of interest, and a detailed detector simulation that describes how detector effects distort the particle property distributions. These distortions affect the reconstruction of kinematic quantities of particles incident to the detector, resulting in an altered observed particle distribution $f_{\text{det}}(\mathbf{y})$, where $\mathbf{y}$ represents the reconstructed detector-level kinematic properties. This relationship between an observed distribution and its underlying kinematic properties can be written as a Fredholm integral equation of the first kind,

$$f_{\text{det}}(\mathbf{y}) = \int d\mathbf{x}\, P(\mathbf{y}|\mathbf{x})\, f_{\text{true}}(\mathbf{x})\,, \tag{1}$$

where $P(\mathbf{y}|\mathbf{x})$ is the conditional probability distribution describing the detector effects. Unfolding requires the inverse process $P(\mathbf{x}|\mathbf{y})$, which can be expressed with Bayes' theorem, as

$$P(\mathbf{x}|\mathbf{y}) = \frac{P(\mathbf{y}|\mathbf{x})f_{\text{true}}(\mathbf{x})}{f_{\text{det}}(\mathbf{y})}\,. \tag{2}$$

In this context, a detector dataset can be unfolded by sampling from the posterior $P(\mathbf{x}|\mathbf{y})$ to recover the distribution $f_{\text{true}}(\mathbf{x})$. The detector effects $P(\mathbf{y}|\mathbf{x})$ are assumed to be the same for any physics process, and it is clear that the posterior $P(\mathbf{x}|\mathbf{y})$ depends on the prior distribution $f_{\text{true}}(\mathbf{x})$. Although one can sample from $f_{\text{true}}(\mathbf{x})$ through the use of particle generators, there is no guarantee that any particular assumed $f_{\text{true}}(\mathbf{x})$ accurately represents the underlying physics of the specific data to unfold. Consequently, unfolding results can be significantly influenced

by the assumed underlying distribution, potentially introducing bias or limiting the method's ability to detect unexpected phenomena, a challenge commonly known as the bias problem in unfolding. Although many traditional unfolding methods [2–4] attempt to address this bias problem, they face several inherent limitations: they require binned histograms, cannot unfold multiple observables simultaneously, retain a potential residual bias toward an assumed underlying distribution (introduced as a necessary trade-off to reduce large variances in the unfolded distribution) and depend heavily on specific choices made by experimenters during the data analysis process. These challenges reveal the difficulty in developing a *universal* unfolder, which aims to remove detector effects from any set of measured data agnostic of the process of interest with no bias towards any prior distribution and without constraints on the final interpretation. This task of creating a widely applicable unfolding method is known as the generalization problem.

**Related work:** Various machine learning approaches have emerged in recent years to address these challenges. These include re-weighting methods like OmniFold [5, 6], as well as several generative approaches. Among generative techniques are those that use conditional invertible neural networks (cINN) [7–9], generative adversarial networks (GAN) [10, 11], and variational latent diffusion models (VLD) [12, 13]. Furthermore, distribution mapping techniques have been developed such as SBUnfold, which utilizes Schrödinger bridges [14], and DiDi, a direct diffusion model [15]. For a complete overview of these methods, see the recent survey by [16]. Each new method has made further strides in unfolding and showed the advantages in machine learning-based approaches compared to traditional techniques. Additionally, complementary studies have explored how training data composition affects ML unfolding performance, including the benefits of augmenting training samples with diverse physics processes to improve phase space coverage and preserve new physics signatures [17]. Table 1 provides a comparative summary of these methods based on machine learning, traditional techniques, and our proposed approach, highlighting some key characteristics and advantages. Although this overview is not exhaustive of all unfolding algorithms developed to date, it provides a comprehensive portrait that allows situating our proposed approach within the landscape of existing solutions to the unfolding problem.

Beyond these methodological developments, machine learning-based unfolding has demonstrated success in various experimental analyses. The ATLAS collaboration has successfully implemented ML unfolding methods, specifically OmniFold, to Z+jets measurements [18], and the LHCb collaboration has applied these methods to jet fragmentation studies [19]. Additionally, the H1 collaboration at HERA has validated applications of ML unfolding in deep-inelastic scattering measurements [20] and jet substructure analyses [21]. These successful applications in experimental analyses demonstrate the viability of machine learning approaches to unfolding.

**Objectives:** Our work seeks to overcome the limitations of traditional unfolding methods while expanding on the benefits offered by machine learning-based approaches. The proposed approach builds upon the advantages of object-wise unfolding, a technique common in machine learning-based unfolding methods, which reconstructs the properties of individual particles or physics objects rather than operating on binned distributions. Through object-wise unfolding, some of the challenges posed by traditional methods can be addressed: the impact of the experimenter's selections and cuts on the unfolded results can be minimized, while underlying correlations between the unfolded distributions are preserved.

We first present a "dedicated" unfolder, an approach similar to many existing machine learning-based methods, which learns and applies a specific posterior distribution for a particular physics process. This approach serves as an effective solution for well-understood pro-

cesses and provides a benchmark for our subsequent work. Building upon this foundation, our aim is to develop a "generalizable" unfolder to handle a wide range of physics processes and observables, including those not explicitly seen during training. This generalization capability is crucial for enhancing the method's applicability across various physics scenarios, while ideally avoiding dependence on specific physics generator models. This amounts to addressing both the bias and generalization problems in our solution to unfolding. Such a method would enable the unfolding of distributions for a wide range of processes, and may extend to some cases involving yet-undiscovered particles in new physics searches at high-energy colliders.

An effective new unfolding method should achieve an accuracy that falls within the typical uncertainty range of measurements where unfolding is applied. For instance, the ATLAS collaboration's [22] measurement of the $W$+jets differential cross-sections [23] obtained from data resulting from proton-proton collisions at the Large Hadron Collider [24] provides a benchmark for the necessary level of precision. These results [25] demonstrate that a 10-15% total uncertainty is typical for energy-momentum related quantities, with approximately 3-5% attributable to unfolding. The goal is to achieve this level of accuracy while simultaneously preserving the benefits of object-wise unfolding, such as maintaining correlations between kinematic quantities, and offering generalization capabilities. With these objectives, we hope to contribute a more flexible, accurate, and widely applicable unfolding tool to the high-energy physics community.

**Our contribution:**   This work introduces a novel approach to unfolding that incorporates statistical moments of distributions as conditioning information within conditional Denoising Diffusion Probabilistic Models (cDDPM) to unfold detector effects in HEP data. We demonstrate that this moment-conditioned approach enables a single cDDPM, trained on diverse particle data, to serve as a "generalized" unfolder by performing multidimensional object-wise unfolding for multiple physics processes without explicit assumptions about the underlying distribution, thereby minimizing bias. We demonstrate our approach using QCD jets, though the method is directly applicable to other physics objects such as leptons and photons.

## 2   Methods

### 2.1   Moment-conditioned generalized unfolding

We seek an approach that will enhance the inductive bias of the unfolding method to improve generalization to cover various posteriors pertaining to different physics data distributions, while avoiding systematically favoring any particular prior distribution. From the a priori information used in the formulation of the solution to the unfolding problem (Equation 2), it can be seen that the posteriors for two different physics processes $i$ and $j$, where the detector effects are independent of the process, are related by a ratio of the probability density functions of each process,

$$\frac{P_i(\mathbf{x}|\mathbf{y})}{P_j(\mathbf{x}|\mathbf{y})} = \frac{f_{\text{true}}^i(\mathbf{x}) f_{\text{det}}^j(\mathbf{y})}{f_{\text{det}}^i(\mathbf{y}) f_{\text{true}}^j(\mathbf{x})}. \tag{3}$$

This relationship indicates that if a posterior for a given physics process can be learned, then distributional information about $f_{\text{true}}(\mathbf{x})$ and $f_{\text{det}}(\mathbf{y})$ could be used to estimate unseen posteriors. Specifically, the first moments of these distributions can be utilized as key features. This approach of using summary statistics like moments is particularly advantageous, as it allows for generalization without being overly sensitive to details and insignificant fluctuations in the distributions. By making use of the first moments of the detector data distribution as conditionals, a more flexible unfolder can be created that is not strictly tied to a selected prior

Table 1: Comparison of unfolding techniques and their key characteristics. The table presents traditional and machine learning-based approaches, with our proposed methods (dedicated and generalizable cDDPM) highlighted in green and orange, respectively. The "Type" column indicates the fundamental algorithmic structure of each method. "Posterior Estimation" describes whether the solution is obtained iteratively, non-iteratively, or partially, where partial refers to methods that estimate only certain components of the posterior. "Event-wise" indicates methods that unfold individual particles or objects without binning, preserving event-level information. This includes both direct event-wise approaches and object-wise methods that can reconstruct complete events by tracking object associations. "Tuneable Regularization" indicates whether the method implements adjustable bias-variance trade-offs in its unfolding solution. "Generalizable" indicates whether the method is designed to unfold diverse physics processes without retraining, using a single model that can handle previously unseen process types through learned inductive biases rather than process-specific training. While not exhaustive, these characteristics provide a framework for comparing different approaches to the unfolding problem.

| | Method | Type | Posterior Estimation | Event-Wise? | Multi-dimensional? | Tuneable Regularization? | Generalizable? |
|---|---|---|---|---|---|---|---|
| Traditional | IBU [2] | Bin-by-bin correction | Iterative | No | Limited | Yes | No |
| | SVD Unfolding [4] | Matrix inversion | Partial | No | Limited | Yes | No |
| | TUnfold [3] | Matrix inversion | Non-iterative | No | No | Yes | No |
| ML-Based | OmniFold [5,6] | Re-weighting | Iterative | Yes | Yes | Yes | No |
| | cINN (iterative) [7] | Generative | Iterative | Yes | Yes | No | No |
| | cINN [8,9] | Generative | Non-iterative | Yes | Yes | No | No |
| | GANs [10,11] | Generative | Non-iterative | Yes | Yes | No | No |
| | VLD [12,13] | Generative | Non-iterative | Yes | Yes | No | No |
| | SBUnfold [14] | Distribution mapping | Non-iterative | Yes | Yes | No | No |
| | DiDi [15] | Distribution mapping | Non-iterative | Yes | Yes | No | No |
| | Dedicated cDDPM | Generative | Non-iterative | Yes | Yes | No | No |
| | Generalizable cDDPM | Generative | Non-iterative | Yes | Yes | No | Yes |

distribution, and enables interpolation and extrapolation to unseen posteriors based on the provided moments. Consequently, this unfolding tool gains the ability to handle a wider range of physics processes and enhances the generalization capabilities, making it a more versatile tool for unfolding in various high energy physics applications. It would therefore provide an unfolding solution addressing both the bias and generalization problems. Crucially, this moment-based conditioning approach requires a generative model that can directly condition on the provided information without introducing explicit bias from training priors, as discussed in Section 2.2.

## 2.2 Denoising diffusion probabilistic models

In learning systems, the challenge of generalization through inductive bias is central, as any system must have some bias beyond the training instances to make the inductive leap necessary to classify unseen cases. Our proposed unfolding approach calls for a flexible generative model to address this challenge, and denoising diffusion probabilistic models (DDPMs) [26] lend themselves naturally to this task. DDPMs are designed to create new content based on training data, making them well-suited for these generalization needs. DDPMs can be trained to model a data distribution through a reversible generative process, which can be conditioned directly on the detector data values and on the moments of the distribution $f_{\text{det}}(\mathbf{y})$. This learned process provides a natural way to sample from $P(\mathbf{x}|\mathbf{y})$ for unfolding. We will first describe DDPMs in a general context before discussing their specific application to our unfolding problem.

**Unconditional DDPM** The standard unconditional DDPM [26] consists of two parts. First is a forward process (or diffusion process) $q(\mathbf{x}_t|\mathbf{x}_{t-1})$ which is fixed to a Markov chain that gradually adds Gaussian noise (following a variance schedule $\beta_1, ..., \beta_T$) to data samples from a known initial distribution,

$$q(\mathbf{x}_t|\mathbf{x}_{t-1}) := \mathcal{N}\left(\mathbf{x}_t\,;\sqrt{1-\beta_t}\,\mathbf{x}_t,\,\beta_t\,\mathbf{I}\right). \tag{4}$$

Second is a learned reverse process (or denoising process) $p_\theta(\mathbf{x}_{0:T})$ parameterized by $\theta$. The reverse process is also a Markov chain with learned Gaussian transitions starting at $p(\mathbf{x}_T) = \mathcal{N}(\mathbf{x}_T; \mathbf{0}, \mathbf{I})$,

$$p_\theta(\mathbf{x}_{0:T}) := p(\mathbf{x}_T)\prod_{t=1}^{T} p_\theta(\mathbf{x}_{t-1}|\mathbf{x}_t), \tag{5}$$

$$p_\theta(\mathbf{x}_{t-1}|\mathbf{x}_t) := \mathcal{N}\left(\mathbf{x}_{t-1}\,;\boldsymbol{\mu}_\theta(t,\mathbf{x}_t),\,\sigma_t^2\mathbf{I}\right). \tag{6}$$

By learning to reverse the forward diffusion process, the model learns meaningful latent representations of the underlying data and is able to remove noise from data to generate new samples from the associated data distribution. This type of generative model has natural applications in high energy physics, for example generating data samples from known particle distributions. However, to be used in unfolding the process must be altered so that the denoising procedure is dependent on the observed detector data, $\mathbf{y}$. This dependence on the observed data is crucial because the goal of unfolding is to reconstruct the true particle-level properties from the observed detector-level data, necessitating a direct link between the denoising process and the specific detector measurements. This can be achieved by incorporating conditioning methods to the DDPM.

**Conditional DDPM** Conditioning methods for DDPMs can either use conditions to guide unconditional DDPMs in the reverse process [27], or they can incorporate direct conditions to the learned reverse process. While guided diffusion methods have had great success in image synthesis [28], direct conditioning provides a framework that is particularly useful in unfolding since it allows for a more explicit and precise incorporation of the detector-level data into the unfolding process, enabling the model to learn a direct mapping between the observed detector measurements and the true particle properties.

We implement a conditional DDPM (cDDPM) for unfolding that keeps the original unconditional forward process and introduces a simple, direct conditioning on the input $\mathbf{y}$ to the reverse process,

$$p_\theta(\mathbf{x}_{0:T}|\mathbf{y}) := p(\mathbf{x}_T|\mathbf{y})\prod_{t=1}^{T} p_\theta(\mathbf{x}_{t-1}|\mathbf{x}_t,\mathbf{y}). \tag{7}$$

Similar to an unconditional DDPM, this reverse process has the same functional form as the forward process and can be expressed as a Gaussian transition with a learned mean $\boldsymbol{\mu}_\theta$ and a fixed variance at each timestep $\sigma_t^2$,

$$p_\theta(\mathbf{x}_{t-1}|\mathbf{x}_t,\mathbf{y}) := \mathcal{N}\left(\mathbf{x}_{t-1}\,;\boldsymbol{\mu}_\theta(t,\mathbf{x}_t,\mathbf{y}),\,\sigma_t^2\mathbf{I}\right). \tag{8}$$

This conditioned reverse process learns to model the posterior probability $P(\mathbf{x}|\mathbf{y})$ through its Gaussian transitions. The reverse process $p_\theta(\mathbf{x}_{0:T}|\mathbf{y})$ is parameterized by $\theta$, where $\theta$ represents the learnable parameters of the model, such as the weights and biases of a neural network. This process learns to remove the noise introduced during the forward process to recover the target $\mathbf{x}$ by conditioning directly on the input $\mathbf{y}$.

Training optimizes the parameters $\theta$ to maximize the likelihood of accurately estimating the noise $\epsilon$ that should be removed at each timestep in order to denoise $\mathbf{x}_t$ given the condition $\mathbf{y}$. Similar to the unconditional DDPM, the Gaussian nature of these transitions and a reparametrization of the mean can be used to simplify the loss function to the mean squared error (MSE) between the noise $\epsilon$ added at each timestep during the forward process and the noise $\epsilon_\theta(\mathbf{x}_t, \mathbf{y})$ predicted by the model given the noisy sample $\mathbf{x}_t$ at timestep $t$ and the condition $\mathbf{y}$:

$$L(\theta) = \mathbb{E}_{t, \epsilon, \mathbf{x}_t, \mathbf{y}} \left[ \left\| \epsilon - \epsilon_\theta\left(t, \mathbf{x}_t, \mathbf{y}\right) \right\|^2 \right]. \tag{9}$$

A detailed derivation of this loss can be found in A.1. This approach can be compared to the commonly used guided conditioning method, where the model estimates the noise with a weighted combination of the conditional prediction and the unconditional prediction as $\tilde{\epsilon}_\theta(\mathbf{x}_t, \mathbf{y}) = (1 + w)\epsilon_\theta(\mathbf{x}_t, \mathbf{y}) - w\epsilon_\theta(\mathbf{x}_t)$ [29]. In the guided approach, the use of the unconditional prediction $\epsilon_\theta(\mathbf{x}_t)$ introduces a bias towards the underlying distribution used in training. This bias can prove detrimental for the unfolding task, as it is important to minimize any assumptions about or dependence on the underlying distribution. The cDDPM approach mitigates this bias by setting the guidance weight $w = 0$, relying solely on the conditional prediction. This choice is particularly critical for our moment-conditioning approach: guided conditioning would contaminate the moment-based generalization by introducing bias from the mixed training dataset, preventing the model from properly interpolating to unseen distributions based solely on the provided moments. In the implementation of the cDDPM, sampling would be done purely according to the learned conditional distribution $p_\theta(\mathbf{x}_t|\mathbf{y})$. Although the learned conditional probability implicitly depends on the prior, sampling from the cDDPM does not require explicitly evaluating the prior distribution $p(\mathbf{x}_t)$ over the data space. This makes the cDDPM a promising choice for applications like unfolding where the prior is unknown or difficult to model.

Our moment-conditioned cDDPM approach differs fundamentally from other diffusion-based unfolding methods in both architecture and conditioning strategy. Variational latent diffusion models (VLD) [12, 13] operate in a compressed latent space and utilize additional encoder-decoder networks. DiDi [15] employs direct diffusion that maps between detector and truth distributions without incorporating distributional conditioning information. Generally, other existing diffusion-based unfolders typically employ guided conditioning or operate on single physics processes, preventing them from achieving the generalization capabilities central to our approach. Our implementation of direct conditioning combined with moment-based generalization represents a novel combination that enables unfolding across multiple physics processes without retraining.

## 2.3 Unfolding with cDDPMs: Toy model

Proof-of-concept is first demonstrated using a toy model with non-physics data. The method will then be tested with generated physics data, discussed in Section 3.

In both the toy model and physics results, the Wasserstein-1 distance [30] is used to measure the success of the proposed unfolding algorithm. This metric quantifies the discrepancy between two distributions, quantifying how closely the unfolded distribution matches the target (truth) distribution, as well as the difference between the detector data and the true underlying distribution. The objects to unfold are characterized by multiple kinematic properties, necessitating a multidimensional representation of each object. Unlike traditional methods that unfold each quantity separately thereby losing correlations, the cDDPM algorithm preserves these correlations by simultaneously unfolding the full multidimensional phase space. The 1D Wasserstein distances for individual quantities are provided (labeled "Wasserstein" in

figures), as well as multidimensional Wasserstein distances for the complete set of variables defining physics objects (reported in tables).

In addition to the Wasserstein distance, two other metrics are employed to evaluate the unfolding performance. The first is the chi-squared per degree of freedom ($\chi^2$/DoF) on the binned distribution, denoted as "Binned $\chi^2$/DoF" in the figures. This metric assesses the agreement between the unfolded and true distributions while accounting for statistical fluctuations in each bin. The second metric is the sum of the absolute values of (ratios - 1), where the ratios are calculated as the detector or unfolded distributions divided by the truth distribution. This metric, labeled as "$\sum |\text{ratios} - 1|$" in the figures, provides a measure of the overall deviation from the true distribution across all bins.

In the figures, the binning of the distributions is done after unfolding to provide a simple visual representation of the results. From the unfolded results, any choice of binning can be used, allowing the data to be presented in various ways and adapted for specific physics analyses.

This study focuses on QCD jets, which are narrow streams of hadrons produced by quark or gluon hadronization in high-energy particle collisions. Jets are typical object signatures in HEP data that provide information about the fundamental interaction of nature that leads to their production. Multiple toy model jet datasets are designed, each representing a distinct physics process, with each dataset independently distorted by detector effects. Each jet is characterized by a 4-vector containing kinematic information: transverse momentum ($p_T$), pseudorapidity ($\eta$), azimuthal angle ($\phi$), and energy ($E$). To emulate realistic physics data, these parameters are each sampled from specific distributions. The particle $p_T$ is sampled from an exponential distribution $f(x; 1/\beta) = (1/\beta) \exp(-x/\beta)$, reflecting the typical exponential behavior observed in $p_T$ distributions in particle physics. The azimuthal angle $\phi$ is sampled uniformly from the range $[-\pi, \pi]$, while the pseudorapidity $\eta$ follows a Gaussian distribution with $\mu = 0$ and $\sigma = 2$. Assuming massless jets for simplicity, the energy is calculated as $E = p_T \cosh \eta$. These components for the "truth-level" jet vector ($\mathbf{x}$), which is then processed through a detector-like smearing framework, producing a "detector-level" jet quantities ($\mathbf{y}$) that mimics the particle interactions within an actual detector. For a comprehensive description of the detector smearing, please refer to B.3.

**Part 1: Dedicated unfolder**  We first consider how to setup a *dedicated* cDDPM unfolder (without use of the distributional moments) that can achieve multidimensional object-wise unfolding for a single physics process. A cDDPM can be trained with data pairs ($\mathbf{x}, \mathbf{y}$) as input to learn the posterior distribution $P(\mathbf{x}|\mathbf{y})$. To unfold, the detector data $\mathbf{y}$ are given as input and the cDDPM acts as a posterior sampler of $P(\mathbf{x}|\mathbf{y})$. This dedicated unfolder relies on learning a specific posterior distribution, which implicitly incorporates information about the prior distribution of the training data. While not explicitly using the prior, this approach results in an unfolder that is more tailored to the particular distribution represented in the training data. This makes the dedicated unfolder particularly well-suited for scenarios involving well-known distributions, where the implicit bias towards the training prior is acceptable or even desired.

To validate this approach, two test cases are presented that probe different aspects of the cDDPM dedicated unfolder. In case (1), the ability of the cDDPM to learn a posterior $P(\mathbf{x}|\mathbf{y})$ given a dataset of pairs $\{\mathbf{x}, \mathbf{y}\}$ is tested, where both the training and test datasets have the same prior and posterior distributions (Figure 1a). Case (2), depicted in Figure 1b, evaluates the unfolding accuracy when the training and test datasets have different underlying true distributions (priors) but the exact same posterior $P(\mathbf{x}|\mathbf{y})$. Since the cDDPM does not explicitly evaluate the prior distribution of the training dataset, it can sample from the posterior distribution without an imposed bias towards underlying characteristics of the prior of the training data. The successful unfolding in each case validates the cDDPM formulation, showing that is

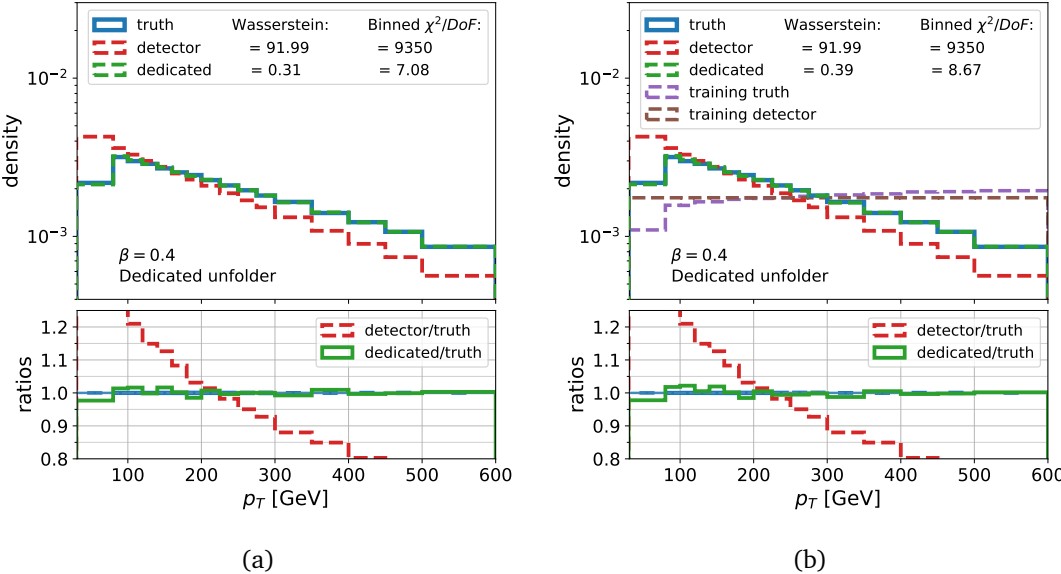

Figure 1: Unfolding results for toy-model data using a cDDPM dedicated unfolder. On the left (a) are the results for case (1) that tests the ability of the cDDPM to learn a posterior $P(\mathbf{x}|\mathbf{y})$ given a dataset of pairs $\{\mathbf{x}, \mathbf{y}\}$. On the right (b) are the results for case (2) which aims to unfolds the same test dataset (red) as (a), but the cDDPM unfolder is trained using an alternative training dataset (purple and brown) that is created such that the posterior $P(\mathbf{x}|\mathbf{y})$ is the same as test data, while the priors (blue vs purple) are different.

able to learn the posterior $P(\mathbf{x}|\mathbf{y})$ with minimal influence from the specific shape of the training distribution, a crucial feature for developing our unbiased generalized unfolder.

**Part 2: Generalized unfolder**    Since physics generators cannot perfectly emulate real physics processes, it is important to minimize assumptions and bias towards specific generator models. Although the dedicated unfolder exhibits reduced bias towards the underlying distribution used in training, the learned posterior by definition contains information about the prior. Consequently, the posterior will not be the same across different data distributions subjected to the same detector effects. This means the generalization power of the dedicated unfolder is limited, as it is strictly tied to one specific posterior, and it may not successfully unfold data for which the observed distribution widely differs from those assumed in the training phase of the algorithm. Here, the aim is to develop a *generalized* cDDPM unfolder capable of handling a broader range of posteriors, thus enabling the unfolding of data from diverse physics processes with minimal bias towards generated physics distributions.

To achieve this, the model's inductive bias is enhanced by incorporating distributional moments into the approach, allowing for interpolation and extrapolation to unseen posteriors as discussed in Section 2.1. This strategy is implemented by expanding the training dataset to include data pairs $(\mathbf{x}, \mathbf{y})$ from multiple different distributions. For each dataset, six moments of the $p_T$ distribution are computed: the first raw moment (mean) $\mu = \frac{1}{N} \sum_{i=1}^{N} p_{T,i}$, followed by the 2nd through 6th central moments calculated as $\mu_k = \frac{1}{N} \sum_{i=1}^{N} (p_{T,i} - \mu)^k$ for $k = 2, ..., 6$. These moments are process-specific – they are calculated once for each physics process using the full set of events for that process. The moments are then appended to both the truth-level ($\mathbf{x}$) and detector-level ($\mathbf{y}$) vectors of every jet from that process. This means that while the kinematic components of $\mathbf{x}$ and $\mathbf{y}$ vary jet-by-jet, all jets from the same physics process

carry identical moment values that characterize that process's $p_T$ distribution. The resulting augmented vectors contain both the jet-specific kinematic information and these process-level distributional features that help distinguish between different physics processes. During evaluation on real experimental data, the six moments are calculated directly from the entire dataset to be unfolded, regardless of the underlying physics composition. The trained model then uses these observed moments to determine the appropriate posterior for unfolding, without requiring prior knowledge of which physics processes are present in the data.

The moments of the $p_T$ distribution are chosen because they are characteristic features for different physics processes, with the raw first moment capturing the average scale of the process and the higher central moments characterizing the shape and asymmetry of the distribution. Tests were performed incorporating moments from all components of the jet kinematic vector ($p_T$, $\eta$, $\phi$, $E$), but this resulted in degraded performance. This degradation likely stems from the inclusion of moments from components like $\eta$ and $\phi$ distributions that remain similar across different physics processes, effectively introducing noise to the conditioning information rather than providing discriminating features. Additional tests comparing the use of four versus six moments showed improved performance with six moments, leading to this choice for the final implementation, though the possibility of including even higher moments was not investigated.

Two approaches were explored for incorporating the distributional moments into the unfolding process. The first method appends the moments only to the detector-level vector $\mathbf{y}$, using them purely as conditioning information. The second approach includes the moments in both the truth-level and detector-level vectors ($\mathbf{x}$ and $\mathbf{y}$, respectively), allowing the truth-level moments to participate in the denoising process. While both approaches showed promising results, the latter demonstrated marginally better performance and was therefore adopted for all results presented in this work. This improvement may occur because including moments in both vectors creates consistency constraints during training. The model must ensure that denoised kinematic properties align with denoised distributional moments, providing additional guidance that improves the overall accuracy of the reconstruction.

With this chosen approach, the truth-level vector $\mathbf{x}$ and detector-level vector $\mathbf{y}$ are now redefined to include these distributional moments, creating augmented jet vectors that encompass both the original kinematic information and the newly added moment data. It is important to note that while these moments are unfolded along with the jet kinematic components, they primarily serve as conditioning information and are not part of the final output, being discarded after the unfolding process. By training with these diverse augmented data pairs ($\mathbf{x}, \mathbf{y}$), the cDDPM is enabled to represent multiple posteriors corresponding to the distributions in the expanded training dataset, distinguishable through the added distributional information provided by the moments (more details on the training dataset are provided in B.1).

To evaluate the efficacy of the generalized unfolder approach, a series of tests are conducted using a cDDPM trained on an expanded dataset. This dataset incorporates four distinct $p_T$ distributions: a uniform distribution and exponential functions with $\beta = 0.7, 0.3$, and $0.07$, all augmented with their respective distributional moments. The model's performance is then assessed on test datasets sampled from previously unseen $p_T$ distributions ($\beta = 0.4, 0.2, 0.1$, and $0.06$). Figure 2 presents these results, demonstrating the unfolder's ability to generalize across different distributions.

To further investigate the impact of including the distributional moments in the data vector and to disentangle this effect from simply training on diverse data, additional experiments are performed where all models use identical training datasets but with different conditioning approaches. Figure 3 illustrates two critical test cases for comparison: (a) unfolding without including any moments in the training or test datasets, and (b) unfolding with "fake" moments

Figure 2: Unfolding results for toy-model data using a cDDPM generalized unfolder. The unfolder is trained on an expanded dataset including four distinct $p_T$ distributions (uniform and exponential with $\beta = 0.7, 0.3,$ and $0.07$). Panels show unfolding performance on previously unseen $p_T$ distributions: (a) $\beta = 0.4$, (b) $\beta = 0.2$, (c) $\beta = 0.1$, and (d) $\beta = 0.06$. This shows the unfolder's ability to generalize across different distributions, with the unfolded results (orange) closely matching the truth-level data (blue). Error bars indicate statistical uncertainties after binning.

(random numbers) assigned to the distributions. The stark contrast in performance between these cases and the primary results highlights the crucial role that the distributional moments play in achieving a generalized unfolder. These results demonstrate the strong potential use of the cDDPM for generalized unfolding in HEP physics analyses.

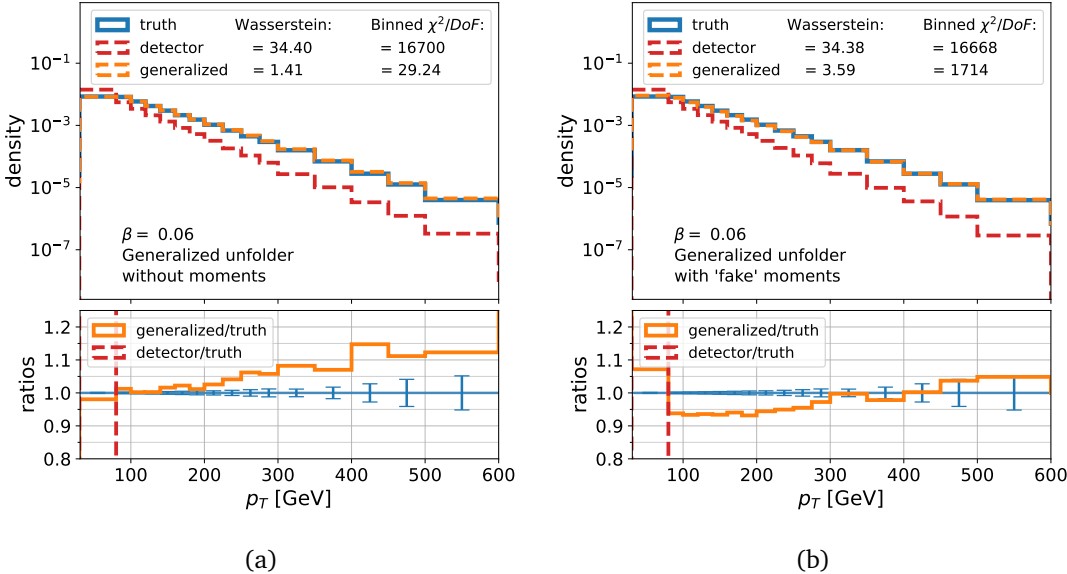

(a)                                          (b)

Figure 3: Investigation of the role of distributional moments in unfolding performance. Using the same distributions as Figure 2 (a uniform distribution as well as exponential distributions with $\beta = 0.7, 0.3, 0.07$) for training and $\beta = 0.06$ for testing, two test cases are shown: (a) unfolding without including any distributional moments in either training or test datasets, and (b) unfolding with random numbers assigned as "fake" moments. The degraded performance in both cases, compared to Figure 2c, demonstrates the crucial role that true distributional moments play in achieving successful generalized unfolding.

## 3 Application to particle physics data

### 3.1 Setup and implementation

The approach is now tested on simulated particle physics data. The objective is to test whether the method can succeed in accurately unfolding physically relevant observable distributions, rather than simple functions, with more complex detector effect models. Using the PYTHIA event generator [31], jet datasets for various physics processes ($t\bar{t}$, $W$+jets, $Z$+jets, dijet, and leptoquark) are generated under different theoretical modelings of the processes (details of these synthetic datasets can be found in B.2). In this context, the jet kinematic information is defined with a vector that includes the transverse momentum ($p_T$), pseudorapidity ($\eta$), azimuthal angle ($\phi$), and its 4-momentum ($E, p_x, p_y, p_z$).

These jet vectors are defined both at truth-level as $\mathbf{x}$ and detector-level as $\mathbf{y}$, with one vector pair ($\mathbf{x}, \mathbf{y}$) corresponding to each individual jet in an event. The generated truth-level jets were passed through two different detector simulation frameworks to simulate particle interactions within an LHC detector. The detector simulations used were DELPHES [32] with the standard configuration for the CMS detector [33], and another detector smearing framework developed using an analytical data-driven approximation for the $p_T$, $\eta$, and $\phi$ resolutions from results published by the ATLAS collaboration [34] (more details in B.3). DELPHES provides a comprehensive detector simulation that takes into account the full detector geometry and its impact on particle reconstruction, while the data-driven detector smearing focuses on resolution effects, allowing testing of the unfolding success under more drastic detector smearing. Separate generalized unfolders were trained for the datasets from each detector-effects framework, but the implementation and application of the cDDPM methodology remained identical in both cases.

Using the same approach described for the toy model, both dedicated and generalized cD-DPM unfolders are trained. The dedicated unfolder is trained using data pairs $(\mathbf{x}, \mathbf{y})$, excluding the distributional moments. In contrast, the generalized unfolder is trained on multiple simulated physics processes (detailed in B.2). For the data-driven detector smearing framework, the training dataset includes 18 different physics processes, while for the DELPHES framework, 6 different processes were available for training. For each process, the training dataset incorporates the first 6 central moments of the $p_T$ distribution, appending these moments to the corresponding truth-level and detector-level data vectors of that distribution. All results presented in this section use these trained generalized unfolders (one for each detector simulation framework), with all test cases shown being from datasets that were excluded from the training. A key distinction between the generalized and dedicated unfolders lies in their learning outcomes: the generalized unfolder learns to model multiple posteriors from the diverse physics processes in its training data, whereas the dedicated unfolder captures only a single posterior represented by its specific training set. This difference allows us to use the dedicated unfolder as a performance benchmark, against which we can evaluate the effectiveness of the generalized unfolder.

## 3.2 Performance evaluation

Figure 4 showcases two critical test cases that demonstrate the versatility of the generalized unfolder. Panel (a) presents results from an "unknown" process dataset, created by combining jets from multiple sources: 40% $t\bar{t}$, 35% $W$+jets, and 25% leptoquark test datasets. While this "unknown" dataset is constructed from known physics processes (though none were included in the training data), their combination produces a unique prior distribution. The moments used for conditioning are calculated from the combined dataset as a whole, presenting the generalized unfolder with previously unseen distributional characteristics. This scenario represents the optimal use case for the generalized unfolder, as it simulates a situation where the underlying physics is not fully known or understood, such as in new physics searches. To demonstrate this, the generalized unfolder is compared against a dedicated unfolder trained on $t\bar{t}$ data, chosen as it represents the majority component (40%) of the unknown process. The generalized unfolder demonstrates superior performance when unfolding this unknown process, effectively adapting to the mixed nature of the data without prior knowledge of its composition. In contrast, the dedicated unfolder, constrained by its assumption of a $t\bar{t}$-like posterior, shows reduced accuracy. This comparison underscores the generalized unfolder's potential in scenarios involving new or unexpected physics processes, where the underlying distribution may deviate significantly from known models. Panel (b) further demonstrates the generalized unfolder's performance, unfolding data from graviton production in the context of large extra-dimension scenarios [35], accompanied by jets. This process, which features distinctly different physics signatures from Standard Model processes, was completely absent from the training data yet is accurately unfolded by our method. Here too, the generalized unfolder demonstrates accuracy in reconstructing the true distributions, effectively adapting to entirely new physics processes without prior knowledge of their underlying physics. In both cases, the generalized unfolder achieves accuracy within typical LHC uncertainty budgets. This flexibility demonstrated by the generalized unfolder is beneficial for new physics searches and studying processes not accurately modeled by current theories, providing an unfolding solution to the bulk of the data analyses performed at high-energy colliders.

Figures 5 and 6 illustrate unfolding results for the data-driven detecter smearing and DELPHES detector simulation, respectively. The plots show various jet kinematics across different generated physics datasets that are not included in the training data, showcasing the generalized unfolder's versatility. While the generalized unfolder's advantage is expected for unknown processes, we also aim for comparable performance to dedicated unfolders on known

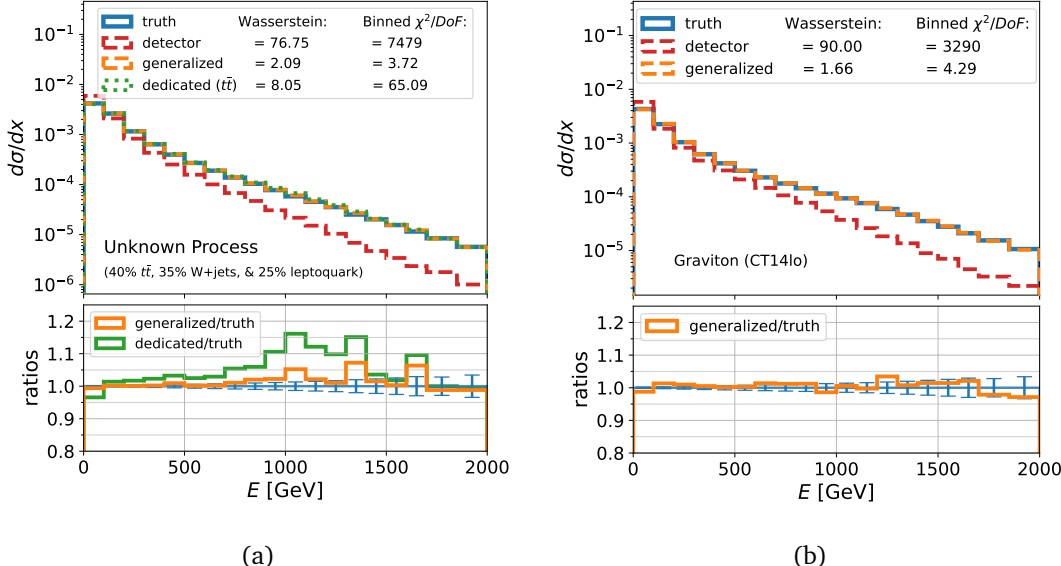

(a)                                                (b)

Figure 4: Unfolding results from the data-driven detector smearing using the generalized cDDPM unfolder. Panel (a) shows performance on data from an "unknown" physics process combining multiple processes. The generalized unfolder (orange) demonstrates superior performance compared to the dedicated unfolder (green), which was trained assuming a specific physics process. Panel (b) shows the generalized unfolder successfully handling data from graviton production accompanied by jets, a new physics process completely absent from the training data. The accuracy of the generalized approach in both scenarios illustrates its ability to handle previously unseen physics processes without assuming an underlying distribution.

processes. The results demonstrate that the performance between the dedicated and generalized unfolders is indeed comparable. The statistical measures of agreement between the unfolded and true distributions slightly favor the dedicated unfolder, particularly in the binned $\chi^2$/DoF metric, due to small fluctuations in the first three bins of the distributions where the statistical uncertainty is negligible. This is particularly evident in the W+jets unfolding (Figure 5c), where the generalized unfolder exhibits nearly perfect agreement with the true distribution in the low-$p_T$ region, achieving better performance than the dedicated unfolder in these bins, despite showing some deviations in the tail of the distribution. When considered in the context of total systematic uncertainties typical in real measurement processes, the observed deviations between dedicated and generalized unfolders would contribute negligibly to the overall measurement uncertainty due to unfolding. Therefore, the generalized unfolder achieves a similar level of accuracy to the dedicated unfolder on known processes, while maintaining the advantage of being able to unfold unknown processes.

To validate our framework's effectiveness we compare both unfolders across various test datasets, and Table 2 presents the resulting multidimensional Wasserstein distances to their true distributions. These quantitative results support the earlier conclusions, showing comparable performance between the generalized and dedicated unfolders across all test processes.

To further evaluate the generalized unfolder's capabilities, a test is conducted using $t\bar{t}$ datasets generated with different theoretical modeling from varying settings in PYTHIA. Dedicated unfolders are trained on each of these $t\bar{t}$ variants, except for one "unseen" variant. The variants used for training the dedicated unfolders employed different PDFs (CTEQ6L1 and NNPDF23) with the default PYTHIA parton showers, while the unseen variant used CT14lo PDF with the Vincia parton shower [36]. The performance of these dedicated unfolders and



Figure 5: Unfolding results of the data-driven detector smearing of jet vector components for three different physics processes: $t\bar{t}$ (a) and (b), leptoquark (c), and $W$+jets (d). For each process, results from the generalized cDDPM unfolder (orange) are compared against a process-specific dedicated unfolder (green), where each dedicated unfolder was trained exclusively data from its corresponding process, while the generalized unfolder was trained on a diverse dataset excluding all three test processes. In all cases, the unfolding accuracy is within the expected uncertainty budget typical of experimental measurements of these distributions. Error bars indicate statistical uncertainties.

the generalized unfolder is then compared in unfolding the unseen $t\bar{t}$ variant. Figure 7 illustrates the results of this test. As shown in Figure 7, the generalized unfolder accurately unfolds the data from the unseen $t\bar{t}$ variant, and in some metrics outperforms the dedicated unfolders trained on the other $t\bar{t}$ variants. This result demonstrates the generalized unfolder's ability to capture subtle differences between posterior distributions arising from different generator settings, not just large variations across different physics processes. Such capability suggests

**Figure 6:** Unfolding results of the DELPHES detector simulation of jet vector components for three different physics processes: leptoquark (a) and (b), $t\bar{t}$ (c), and W+jets (d). For each process, results from the generalized cDDPM unfolder (orange) are compared against a process-specific dedicated unfolder (green). Each dedicated unfolder was trained exclusively on its corresponding physics process, while the generalized unfolder was trained on a diverse dataset excluding all three test processes. In all cases, the unfolding accuracy is within the expected uncertainty budget typical of experimental measurements of these distributions. Error bars indicate statistical uncertainties.

that the generalized unfolder could be a valuable tool in refining models for generating known physics processes, as it can adapt to nuanced variations in the underlying distributions without requiring specific training on each variant of PDF or parton shower model.

In Figure 8, the model's efficacy is further demonstrated with two tests: (1) reconstructing jet mass from unfolded results, indicating well-preserved correlations among jet vector components, and (2) reconstructing event-level observables from unfolded quantities, achieved

Table 2: Comparison of Wasserstein distances for detector-level data and unfolded results using the data-driven detector smearing, corresponding to the processes shown in Figures 5 and 6. Values are shown for both generalized and dedicated unfolders across different physics processes.

| Detector | Process | Multidimensional Wasserstein Distances | | |
|---|---|---|---|---|
| | | Detector | Generalized | Dedicated |
| Data-driven | "Unknown" | 28.20 | 0.74 | 2.68 |
| | Graviton (CT14lo) | 31.35 | 0.64 | N/A |
| | $t\bar{t}$ (CT14lo, Vincia) | 26.43 | 0.34 | 0.35 |
| | Leptoquark (NNPDF23) | 32.42 | 0.26 | 0.30 |
| | W+Jets (CT14lo) | 31.09 | 0.54 | 0.52 |
| DELPHES | $t\bar{t}$ (CTEQ6L1) | 1.49 | 0.21 | 0.20 |
| | Leptoquark (CTEQ6L1) | 1.51 | 0.27 | 0.15 |
| | W+Jets (CTEQ6L1) | 2.07 | 0.62 | 0.34 |

by tracking event numbers through object-wise unfolding. The successful reconstruction of jet mass, which is not directly unfolded but derived from the unfolded jet vector components, showcases the method's ability to maintain complex relationships between variables. This preservation of correlations allows for the calculation of various derived quantities post-unfolding, offering the option to construct new observables that are not explicitly part of the original unfolding process.

### 3.3 Computational performance

The generalized unfolder demonstrates not only accuracy but also computational efficiency, making it a practical tool for large-scale physics analyses. The model used in these results, trained on a diverse dataset of 1.8 million jets, requires approximately 3 hours of training time on an NVIDIA A100 GPU. Once trained, the generalized unfolder can be applied rapidly, with the ability to unfold 1 million events in approximately 3 minutes. This speed is particularly advantageous as the generalized unfolder does not require retraining for specific processes, unlike the dedicated unfolder and other machine-learning based unfolding methods. Consequently, it can provide fast, object-wise unfolding results across a wide range of physics analyses without incurring additional training overhead for each new process or dataset. For a more detailed discussion of the computational performance and parameters of the model, see A.2.

## 4 Conclusion

The results presented in this paper demonstrate the generalized cDDPM unfolder can successfully unfold detector effects on particle jets from a variety of physics processes, including those not seen during training. The key feature of this method is its non-iterative and flexible posterior sampling approach, which exhibits a strong inductive bias allowing generalization to unseen processes without explicitly assuming the underlying physics distribution. The generalized unfolder therefore provides a solution to the unfolding problem that addresses both the bias and the generalization challenges, something other approaches to unfolding did not attempt. Additionally, the generalized unfolder is able to accurately reconstruct jet properties and derived quantities like jet mass, even for processes absent from its training data. By preserving correlations between jet vector components, it enables the construction of complex observables post-unfolding, offering new possibilities in data analysis.

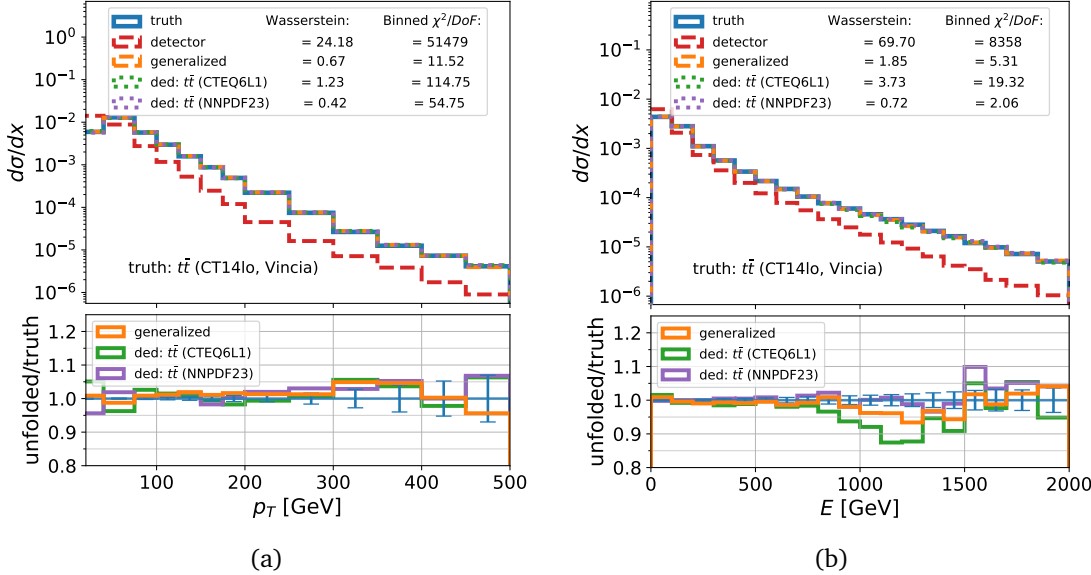

Figure 7: Comparison of unfolding performance for the data-driven detector smearing on an unseen $t\bar{t}$ variant generated using CT14lo PDF and the Vincia parton shower. The generalized unfolder (orange) demonstrates comparable performance to dedicated unfolders trained on other $t\bar{t}$ variants with different PDFs (CTEQ6L1 shown in green, NNPDF23 shown in purple). While these dedicated unfolders were each trained exclusively on their respective $t\bar{t}$ variant, the generalized unfolder was trained on a diverse dataset of multiple physics processes described in the text. Shown are the jet $p_T$ (a) and $E$ (b) distributions.

We note that quantitative comparisons to other ML unfolding methods (OmniFold, GANs, cINNs) would not provide meaningful insights, as these methods are fundamentally designed for different problems. Existing methods require process-specific training and cannot handle unknown physics processes, making direct performance comparisons inappropriate. The key novelty of our moment-conditioning approach lies not in superior performance on known processes, but in enabling the unprecedented capability of process-agnostic unfolding without retraining.

This approach also offers computational advantages. Once trained, it eliminates the need for process-specific retraining, reducing computational overhead. The ability to unfold a million events in approximately 3 minutes demonstrates its potential for efficient large-scale data processing in high-energy physics experiments.

Several open questions remain regarding the implementation of the conditioning on the moments. These include optimal selection of priors and the number of moments required for the best unfolding performance. Further investigation is needed to determine the extent of the cDDPM's inductive bias and its tolerance to variations in the underlying physics processes. Understanding these aspects will help refine the method and ensure its robustness across a wide range of scenarios.

While this approach shows promise, key limitations are acknowledged. The current studies were performed on QCD jets, and extending this method to other particle types is necessary for its comprehensive application in data analysis. Addressing particles outside detector thresholds and accounting for systematic and experimental uncertainties are crucial improvements needed to fully realize the method's potential in practical applications. An important constraint of the current implementation is that while correlations between object vector components are preserved, the model lacks access to event-wise information which impacts the reconstruction

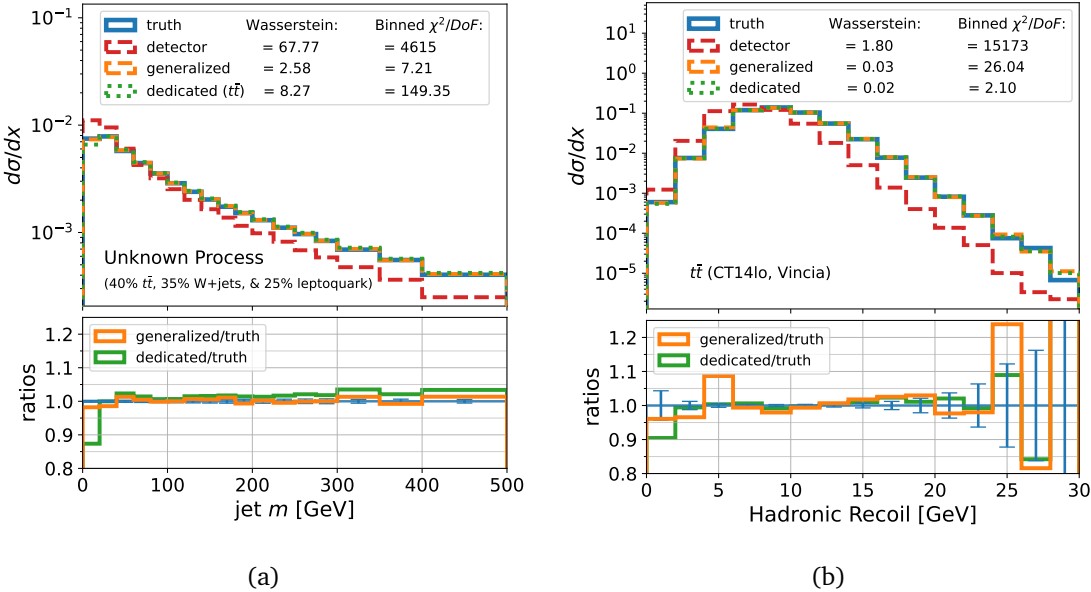

Figure 8: Demonstration of the unfolding method's ability to preserve correlations and handle event-level quantities using the data-driven detector smearing. Reconstruction of jet mass (a), derived from the unfolded four-momentum components, showing that correlations between kinematic variables are well-preserved through the unfolding process. Reconstruction of the hadronic recoil (b), the vector sum of jet momenta in an event, demonstrating the method's capability to handle event-level physics quantities. Error bars indicate statistical uncertainties.

accuracy of certain event-level observables. Additionally, a systematic study of correlation preservation in object-wise unfolding is needed to understand when detector effects factorize at the individual object level. These improvements and extensions are left for future work.

To conclude, the results confirm the versatility of the generalized cDDPM unfolder across diverse physics processes. This non-iterative and flexible posterior sampling approach exhibits a strong inductive bias that allows the cDDPM to generalize to unseen processes without explicitly assuming the underlying distribution, setting it apart from other unfolding techniques developed so far.

# Acknowledgments

**Funding information**    This work has been made possible thanks to the support of the Department of Energy Office of Science through the Grant DE-SC0023964. Shuchin Aeron and Taritree Wonhjirad would also like to acknowledge support by the National Science Foundation under Cooperative Agreement PHY-2019786 (The NSF AI Institute for Artificial Intelligence and Fundamental Interactions, http://iaifi.org/).

**Funding information**    This work was supported by the U.S. Department of Energy, Office of Science, under Award Number DE-SC0023964.

# A  cDDPM details

## A.1  Loss derivation

In the proposed cDDPM, the forward process is a Markov chain that gradually adds Gaussian noise to the data according to a variance schedule $\beta$.

$$q(\mathbf{x}_t|\mathbf{x}_{t-1}) := \mathcal{N}\left(\mathbf{x}_t\,;\sqrt{1-\beta_t}\,\mathbf{x}_{t-1}\,,\,\beta_t\,\mathbf{I}\right). \tag{A.1}$$

To recover the original sample from a Gaussian noise input, this process needs to be reversed. This can be achieved through the use of a model $p_\theta$ which corresponds to the joint distribution $p_\theta(x_{0:T}|y) = p_\theta(x_0, x_1, ... x_T|y)$, and it is defined as a Markov chain with learned Gaussian transitions starting at $p(x_T|y) = \mathcal{N}(x_T; 0, I)$

$$p_\theta\left(\mathbf{x}_{0:T}|\mathbf{y}\right) := p(\mathbf{x}_T|\mathbf{y})\prod_{t=1}^{T} p_\theta\left(\mathbf{x}_{t-1}|\mathbf{x}_t\,,\mathbf{y}\right), \tag{A.2}$$

$$p_\theta(\mathbf{x}_{t-1}|\mathbf{x}_t\,,\mathbf{y}) := \mathcal{N}\left(\mathbf{x}_{t-1}\,;\boldsymbol{\mu}_\theta(t,\mathbf{x}_t,\mathbf{y}),\,\Sigma_\theta(t,\mathbf{x}_t,\mathbf{y})\right), \tag{A.3}$$

where $\mu_\theta$ represents the learned mean, and $\Sigma_\theta$ represents the learned covariance of the Gaussian transitions, which vary with time step $t$:

$$\Sigma_\theta(t,\mathbf{x}_t,\mathbf{y}) = \sigma^2\mathbf{I},\ \sigma^2 = \beta_t\,. \tag{A.4}$$

Training involves learning the reverse Markovian transitions that maximize the likelihood of the training samples, which is equivalent to minimizing the variational upper bound on the negative log likelihood. This negative log likelihood can be expressed in terms of the Kullback-Leibler (KL) divergence [37], a statistical measure of the difference between two probability distributions $P$ and $Q$:

$$D_{KL}(P\|Q) = \sum_{x\in X} P(x)\left(\log\frac{P(x)}{Q(x)}\right). \tag{A.5}$$

Applying this, the variational bound on the negative log likelihood can be expressed as:

$$
\begin{aligned}
\mathbb{E}\big[-\log p_\theta\left(\mathbf{x}_0|\mathbf{y}\right)\big] &\le \mathbb{E}\big[-\log p_\theta\left(\mathbf{x}_0|\mathbf{y}\right)\big] + D_{KL}\left(q(\mathbf{x}_{1:T}|\mathbf{x}_0)\|p_\theta(\mathbf{x}_{1:T}|\mathbf{x}_0,\mathbf{y})\right) \\
&= \mathbb{E}\big[-\log p_\theta\left(\mathbf{x}_0|\mathbf{y}\right)\big] + \mathbb{E}_q\left[\log\frac{q(\mathbf{x}_{1:T}|\mathbf{x}_0)}{p_\theta\left(\mathbf{x}_{1:T}|\mathbf{x}_0,\mathbf{y}\right)}\right] \\
&= \mathbb{E}\big[-\log p_\theta\left(\mathbf{x}_0|\mathbf{y}\right)\big] + \mathbb{E}_q\left[\log\frac{q(\mathbf{x}_{1:T}|\mathbf{x}_0)}{p_\theta\left(\mathbf{x}_{0:T}|\mathbf{y}\right)/p_\theta(\mathbf{x}_0|\mathbf{y})}\right] \\
&= \mathbb{E}\big[-\log p_\theta\left(\mathbf{x}_0|\mathbf{y}\right)\big] + \mathbb{E}_q\left[\log\frac{q(\mathbf{x}_{1:T}|\mathbf{x}_0)}{p_\theta\left(\mathbf{x}_{0:T}|\mathbf{y}\right)}\right] + \mathbb{E}\big[\log p_\theta\left(\mathbf{x}_0|\mathbf{y}\right)\big] \\
&= \mathbb{E}_q\left[-\log\frac{p_\theta(\mathbf{x}_{0:T}|\mathbf{y})}{q(\mathbf{x}_{1:T}|\mathbf{x}_0)}\right] \\
&= \mathbb{E}_q\left[-\log p(\mathbf{x}_T|\mathbf{y}) - \sum_{t\ge 1}\log\frac{p_\theta(\mathbf{x}_{t-1}|\mathbf{x}_t,\mathbf{y})}{q(\mathbf{x}_t|\mathbf{x}_{t-1})}\right] := L\,.
\end{aligned}
\tag{A.6}
$$

Following the similar derivation provided in [26], this loss can then be rewritten using the KL-divergence

$$L = \mathbb{E}_q\left[-\log\frac{p_\theta(\mathbf{x}_{0:T}|\mathbf{y})}{q(\mathbf{x}_{1:T}|\mathbf{x}_0)}\right] \tag{A.7}$$

$$= \mathbb{E}_q\left[-\log p(\mathbf{x}_T|\mathbf{y}) - \sum_{t\geq 1}\log\frac{p_\theta(\mathbf{x}_{t-1}|\mathbf{x}_t,\mathbf{y})}{q(\mathbf{x}_t|\mathbf{x}_{t-1})}\right]$$

$$= \mathbb{E}_q\left[-\log p(\mathbf{x}_T|\mathbf{y}) - \sum_{t>1}\log\frac{p_\theta(\mathbf{x}_{t-1}|\mathbf{x}_t,\mathbf{y})}{q(\mathbf{x}_t|\mathbf{x}_{t-1})} - \log\frac{p_\theta(\mathbf{x}_0|\mathbf{x}_1,\mathbf{y})}{q(\mathbf{x}_1|\mathbf{x}_0)}\right]$$

$$= \mathbb{E}_q\left[-\log p(\mathbf{x}_T|\mathbf{y}) - \sum_{t>1}\log\frac{p_\theta(\mathbf{x}_{t-1}|\mathbf{x}_t,\mathbf{y})}{q(\mathbf{x}_{t-1}|\mathbf{x}_t,\mathbf{x}_0)}\cdot\frac{q(\mathbf{x}_{t-1}|\mathbf{x}_0)}{q(\mathbf{x}_t|\mathbf{x}_0)} - \log\frac{p_\theta(\mathbf{x}_0|\mathbf{x}_1,\mathbf{y})}{q(\mathbf{x}_1|\mathbf{x}_0)}\right]$$

$$= \mathbb{E}_q\left[-\log\frac{p(\mathbf{x}_T|\mathbf{y})}{q(\mathbf{x}_T|\mathbf{x}_0)} - \sum_{t>1}\log\frac{p_\theta(\mathbf{x}_{t-1}|\mathbf{x}_t,\mathbf{y})}{q(\mathbf{x}_{t-1}|\mathbf{x}_t,\mathbf{x}_0)} - \log p_\theta(\mathbf{x}_0|\mathbf{x}_1,\mathbf{y})\right]$$

$$= \mathbb{E}_q\Bigg[\underbrace{D_{KL}\big(q(\mathbf{x}_T|\mathbf{x}_0)\,\|\,p(\mathbf{x}_T|\mathbf{y})\big)}_{L_T} + \sum_{t>1}\underbrace{D_{KL}\big(q(\mathbf{x}_{t-1}|\mathbf{x}_t,\mathbf{x}_0)\,\|\,p_\theta(\mathbf{x}_{t-1}|\mathbf{x}_t,\mathbf{y})\big)}_{L_{1:T-1}} - \underbrace{\log p_\theta(\mathbf{x}_0|\mathbf{x}_1,\mathbf{y})}_{L_0}\Bigg].$$

The term $L_T$ is a constant, as it is the KL-divergence between two distributions of pure noise, and the $L_0$ term is a final denoising step with no comparison to the forward process posteriors. For the term $L_{1:T-1}$, the forward process posteriors can be written as

$$q(\mathbf{x}_{t+1}|\mathbf{x}_t,\mathbf{x}_0) = \mathcal{N}(\mathbf{x}_{t+1};\tilde{\boldsymbol{\mu}}_t(\mathbf{x}_t,\mathbf{x}_0),\tilde{\beta}_t I), \tag{A.8}$$

$$\text{where } \tilde{\beta}_t = \frac{1-\bar{\alpha}_{t-1}}{1-\bar{\alpha}_t}\beta_t,\ \bar{\alpha}_t = \prod_{s=1}^{t}(1-\beta_s),$$

$$\text{and } \tilde{\boldsymbol{\mu}}_t(\mathbf{x}_t,\mathbf{x}_0) = \left(\frac{\beta_t\sqrt{\bar{\alpha}_{t-1}}}{1-\bar{\alpha}_t}\mathbf{x}_0 + \frac{\sqrt{\bar{\alpha}_t}(1-\bar{\alpha}_{t-1})}{(1-\bar{\alpha}_t)}\mathbf{x}_t\right). \tag{A.9}$$

Using this forward process posterior together with the reverse process posterior defined in Equation A.3, a parametrization for $\boldsymbol{\mu}_\theta(\mathbf{x}_t,t,\mathbf{y})$ is introduced that aims to predict $\tilde{\boldsymbol{\mu}}_t(\mathbf{x}_t,\mathbf{x}_0)$. With this the loss becomes

$$L_{t-1} = \mathbb{E}\left[\frac{1}{2\sigma_t^2}\|\tilde{\boldsymbol{\mu}}_t(\mathbf{x}_t,\mathbf{x}_0) - \boldsymbol{\mu}_\theta(t,\mathbf{x}_t,\mathbf{y})\|^2\right] + C, \tag{A.10}$$

where $C$ is a constant, and $\tilde{\boldsymbol{\mu}}_t$ and $\boldsymbol{\mu}_\theta$ can be reparametrized using $\mathbf{x}_t = \sqrt{\bar{\alpha}_t}\mathbf{x}_0 + \sqrt{1-\bar{\alpha}_t}\,\boldsymbol{\epsilon}$ and reduced to

$$L_{t-1} = \mathbb{E}_{\boldsymbol{\epsilon},\mathbf{x}_0,\mathbf{y}}\left[\frac{\beta_t^2}{2\sigma_t^2\alpha_t(1-\bar{\alpha}_t)}\|\boldsymbol{\epsilon} - \boldsymbol{\epsilon}_\theta(t,\sqrt{\bar{\alpha}_t}\mathbf{x}_0 + \sqrt{1-\bar{\alpha}_t}\boldsymbol{\epsilon},\mathbf{y})\|^2\right]. \tag{A.11}$$

Finally we can write a simplified version of the loss with the terms differentiable in $\theta$ as

$$L_{simple}(\theta) = \mathbb{E}_{\boldsymbol{\epsilon},\mathbf{x}_t,\mathbf{y}}\left[\|\boldsymbol{\epsilon} - \boldsymbol{\epsilon}_\theta(t,\sqrt{\bar{\alpha}_t}\mathbf{x}_0 + \sqrt{1-\bar{\alpha}_t}\boldsymbol{\epsilon},\mathbf{y})\|^2\right]$$

$$= \mathbb{E}_{\boldsymbol{\epsilon},\mathbf{x}_t,\mathbf{y}}\left[\|\boldsymbol{\epsilon} - \boldsymbol{\epsilon}_\theta(t,\mathbf{x}_t,\mathbf{y})\|^2\right]. \tag{A.12}$$

This derivation shows that in the cDDPM formulation, the task of learning a posterior distribution reduces to minimizing a simple mean squared error between added and predicted noise. This allows for estimation of the posterior without requiring explicit evaluation of the prior distribution.

---

**Algorithm 1** Conditional DDPM: Training

---

Input: dataset $\{\mathbf{x}_0, \mathbf{y}\}$, variance schedule $\beta_1, \dots \beta_T$

$t \leftarrow \text{Uniform}(\{1, \dots, T\})$

$\bar{\alpha}_t \leftarrow \prod_{s=1}^{t}(1 - \beta_s)$

$\epsilon \leftarrow \mathcal{N}(\mathbf{0}, \mathbf{I})$

**Repeat**

    a)  $\mathbf{x}_t \leftarrow \sqrt{\bar{\alpha}_t}\, \mathbf{x}_0 + \sqrt{1 - \bar{\alpha}_t}\, \epsilon$

    b)  Calculate loss, $L = ||\epsilon - \epsilon_\theta(t, \mathbf{x}_t, \mathbf{y})||^2$

    c)  Update $\theta$ via $\nabla_\theta L$

**Until** converged

---

Figure 9: The training procedure for the conditional DDPM unfolding model is presented. The algorithm trains on data samples $\{\mathbf{x}_0, \mathbf{y}\}$. In step (a) Gaussian noise $\epsilon$ is added to $\mathbf{x}_0$ over $T$ timesteps according to the variance schedule. The model parameterized by $\theta$ is trained to estimate this added noise by observing the noisy states $\mathbf{x}_t$ at a timestep $t$ and the condition $\mathbf{y}$.

## A.2   Model parameters

During inference, the inputs are given to the denoising process are the vector $\mathbf{y}$ and random noise values $\mathbf{x}_T \sim \mathcal{N}(\mathbf{0}, \mathbf{I})$. The denoising process removes noise from $\mathbf{x}_T$ in $T$ steps according to the learned conditional distribution $p_\theta(\mathbf{x}_{0:T}|\mathbf{y})$. Pseudocode for the training and sampling algorithms can be seen in Figures 9 and 10.

    The cDDPM architecture consists of a Multi-Layer Perceptron (MLP), a feedforward neural network, with approximately 1 million trainable parameters. It comprises three main components: an initial linear layer with Gaussian Error Linear Unit (GELU) activation, which provides smooth non-linear transformations, a time step embedding layer, and a series of linear layers with GELU activations. The network takes as input the noised data and the time step. It first processes the input through a 256-unit hidden layer, then adds a learned time step embedding. This combined representation is passed through four 512-unit hidden layers, followed by a 256-unit layer. Skip connections are employed between the input and output of the main block. The final output layer predicts the noise at the given time step. Dropout (rate 0.01) is applied after each linear layer to prevent overfitting during training.

    The diffusion process employs a linear variance schedule over T = 500 time steps. The schedule starts with an initial noise level $\beta_1 = 1e\text{-}4$ at the first step and increases linearly to $\beta_T = 0.02$ at the final step. The model is trained using the Adam optimizer with an initial learning rate of 3e-4. To improve convergence and performance, a linear learning rate scheduler is employed. It starts at the initial rate and linearly decreases to 1% of the initial rate (3e-6) by the end of training.

    The model is trained for 5000 epochs with a batch size of 2048. Using an NVIDIA A100 GPU, the training procedure on our full dataset or 1.8 million data points completes in approximately 3 hours (more details on the training and test datasets in B). Once trained, the model demonstrates efficient inference capabilities. Unfolding a dataset of 1 million data points takes approximately 3 minutes on the A100 GPU, with processing time scaling linearly with the number of jets. Notably, this model functions as a generalizing unfolder, eliminating the need for retraining when applying it to various different datasets.

---

**Algorithm 2** Conditional DDPM: Sampling

---

Input: detector-level data vector $\mathbf{y}$, variance schedule $\beta_1, \ldots \beta_T$

$\mathbf{x}_T \leftarrow \mathcal{N}(\mathbf{0}, \mathbf{I})$

**For** $t = T, \ldots, 1$ **do**

    a) $\alpha_t \leftarrow 1 - \beta_t, \quad \bar{\alpha}_t \leftarrow \prod_{s=1}^{t} \alpha_s, \quad \sigma_t \leftarrow \sqrt{\beta_t}$

    b) $\mathbf{z} \leftarrow \mathcal{N}(\mathbf{0}, \mathbf{I})$ if $t > 1$, else $\mathbf{z} \leftarrow 0$

    c) $\mathbf{x}_{t-1} \leftarrow \frac{1}{\sqrt{\alpha_t}}\left(\mathbf{x}_t - \frac{1 - \alpha_t}{\sqrt{1 - \bar{\alpha}_t}} \boldsymbol{\epsilon}_\theta\left(t, \mathbf{x}_t, \mathbf{y}\right)\right) + \sigma_t \mathbf{z}$

**Return** $\mathbf{x}_0$

---

Figure 10: The trained conditional DDPM model serves as a posterior sampler, generating unfolded truth-level samples $\mathbf{x}_0$ given condition $\mathbf{y}$. Starting from pure noise $\mathbf{x}_T$, the conditioned reverse process denoises $\mathbf{x}_t$ at each timestep by removing the estimated injected noise. Here $\sigma_t \equiv \sqrt{\beta_t}$ since this choice is optimal for a non-deterministic $\mathbf{x}_0$.

## B Datasets

### B.1 Data processing for cDDPMs

The jets in both the toy model and physics datasets are limited to certain ranges for each of the jet vector components, allowing a maximum $p_T, p_x, p_y$ of 1000 GeV and a maximum $p_z$ and $E$ of 4000 GeV, though these ranges can be extended by training with higher energy generated physics datasets. The jet $\eta$ range is between -4.4 and 4.4 (following standard detector limitations), and the $\phi$ range between -3.5 and 3.5. The $\phi$ range is extended beyond $-\pi$ and $\pi$ to avoid discontinuity in training due to wrap-around values. The components of the jet vector are divided by the respective maximum values so that the final vector components each range between $[0,1]$ or $[-1, 1]$. As described in Section 2.3, the first 6 distributional moments of the $p_T$ distribution for each process are appended to the corresponding jet vectors. We choose to use the moments of the $p_T$ distribution because they serve as a distinguishing feature for different physics processes. The $p_T$ spectrum is particularly sensitive to the underlying physics and provides valuable information for process discrimination. We experimented with including moments from all components of the jet vector, but found that this approach led to reduced performance.

    The training dataset for the data-driven detector smearing is comprised of 1.8 million jets generated from 18 different physics simulations described in Table 3. The training dataset for the DELPHES detector simulation is comprised of 1.8 million jets generated from 6 different physics simulations described in Table 4. The data pairs $(\mathbf{x}, \mathbf{y})$ are the input to the training process. The test datasets are processed in the same way as the training dataset, limiting the range and normalizing the range of the values. Additionally, a unique event identifier number, which associates jets to their original event in the dataset, is included in each jet vector and carried through the unfolding process (though not used as an input to the model) to enable the reconstruction of event-level observables after unfolding. The jets in each of test datasets pertain to only one process, with the exception of the "unknown process" dataset which is made from a combination of 3 of the test datasets (40% $t\bar{t}$, 35% $W$+jets, and 25% leptoquarks) to mimic a distribution from an unknown process that could not be easily unfolded with the standard dedicated unfolding approach.

## B.2   Physics generation

Physics datasets are generated using PYTHIA 8.3 Monte-Carlo event generator. The simulations are run for proton-proton collisions at a center-of-mass energy of 13 TeV to emulate LHC physics interactions. The various physics processes used in this study are chosen for their high jet-production cross-section across a large jet energy range. The chosen processes are $t\bar{t}$, $(Z \to \mu\bar{\mu})$+jets, $(W \to \mu\bar{\nu})$+jets, dijets, and a new-physics process of leptoquarks. Note that each listed process typically includes multiple subprocesses (e.g., $t\bar{t}$ production occurs through both gluon-gluon and quark-antiquark collisions). For processes with unstable particles, one particle per event decays to include at least one charged lepton, while other unstable particles decay hadronically. Each of these processes are run under multiple generator settings controlling the theoretical modeling of the underlying processes, with varying parton distribution functions (PDFs), parton shower models, and with an imposed phase space bias that increased the probability of generating events with high jet energies. For simulations where a phase space bias is applied, the events are sampled in the phase space as $(\hat{p}_T/p_T^{\text{ref}})^a$, where we set $p_T^{\text{ref}} = 100$ GeV and $a = 5$ such that events with a $p_T$ over 100 GeV will be oversampled, increasing the event statistics in high-energy regions [31]. A list of the physics processes generated for the data-driven detector smearing framework is shown in Table 3, and a list of the processes generated for the DELPHES detector simluation is shown in Table 4. Unless stated otherwise, the simulations are run with the PYTHIA simple parton shower model and no phase space bias.

## B.3   Detector smearing and jet matching

Our results present the unfolding of detector effects from two different detector smearing frameworks: DELPHES and a data-driven approach. DELPHES is a framework developed for the simulation of multipurpose detectors for physics studies [32]. Specifically, the DELPHES CMS configuration is frequently used as the detector simulation of choice in recent machine-learning based unfolding studies.

For the toy model studies presented in Section 2.3, the detector effects were simulated using the same resolution functions as the data-driven approach described below, applying the same smearing to the kinematic quantities $p_T$, $\phi$, and $\eta$. However, in the toy model case, the jets were treated as massless particles for simplicity, with their energy calculated directly from the smeared momentum components.

To test the unfolding performance under more exaggerated detector effects, we develop a framework with a data-driven detector smearing using jet energy resolution results published by the ATLAS collaboration at a centre-of-mass energy of 8 TeV with an integrated luminosity of 20 fb$^{-1}$ [34]. In this framework, the PYTHIA event generator is used to simulate truth-level particles, and the resulting partons are grouped into jets using the FastJet package [38]. The transverse momentum $p_T$, azimuthal angle $\phi$, and pseudorapidity $\eta$ of each truth-level jet is then smeared following an approximated ATLAS calibration and resolution functions. For the $\phi$ and $\eta$ smearing, the effect is small since the angular resolution effects are proportional to the detector granularity. We assume that there is no angular shift and apply a smearing to $\phi$ and $\eta$ by sampling from a Gaussian centered at the truth-level value and with a $\sigma$ equal to the detector resolution for the particle. We apply a quadratic fit ($\sigma = a\,p_T^2 + b\,p_T + c$) to the calibration data presented in [34] to approximate the detector resolution in $\phi$ and $\eta$.

In principle, a calorimeter cell measurement is an energy measurement, but since the jet calibration studies precisely measure the jet $p_T$ resolution, we apply a shift and a smearing to the jet $p_T$ instead. The jet $p_T$ resolution can be expressed as $\sigma_{p_T} = p_T \sqrt{a/p_T^2 + b/p_T + c}$ and a fit of this function is applied to the jet calibration data (obtained for jets with $0 < |\eta| < 0.8$ for simplicity) to approximate this resolution. The jet $p_T$ also has a calibration shift, which is

Table 3: List of physics simulations generated for the data-driven detector smearing, along with the corresponding parton distribution functions (PDFs), parton shower models, phase space biases, and their inclusion in the training dataset. Simulations not included in the training dataset are used as test datasets.

| Process | PDF with Parton Shower (Phase Space Bias) | In Training? |
|---|---|---|
| $t\bar{t}$ | CT14lo | ✓ |
| | CT14lo (biased) | ✓ |
| | CT14lo with Vincia | |
| | NNPDF23_lo | ✓ |
| | CTEQ6L1 | ✓ |
| | CTEQ6L1 (biased) | ✓ |
| $Z$+jets | CT14lo | ✓ |
| | CT14lo (biased) | ✓ |
| | NNPDF23_lo | ✓ |
| | CTEQ6L1 | |
| | CTEQ6L1 (biased) | ✓ |
| $W$+jets | CT14lo | |
| | CT14lo (biased) | ✓ |
| | NNPDF23_lo | ✓ |
| | CTEQ6L1 | ✓ |
| Dijets | CT14lo | ✓ |
| | CTEQ6L1 | ✓ |
| | CTEQ6L1 (biased) | ✓ |
| Leptoquark | CT14lo | ✓ |
| | CT14lo (biased) | ✓ |
| | NNPDF23_lo | |
| | CTEQ6L1 | ✓ |

calculated from the data. The detector smeared $p_T$ is then defined by sampling from a Gaussian centered at the shifted $p_T$ and with $\sigma$ equal to the jet $p_T$ resolution. Finally, the smeared energy for each jet is calculated with $E = \sqrt{m^2 + |\vec{p}\,|^2}$ by fixing the mass of the particle $m$ and using the smeared $p_T$. This approach enables testing the unfolding algorithm's performance under various detector resolution conditions, including unrealistically large smearing effects. While each detector model requires retraining the unfolder, the algorithm's performance characteristics remain consistent across different detector modelings.

Table 4: List of physics simulations generated for the DELPHES CMS detector simulation, along with the corresponding parton distribution functions (PDFs), phase space biases, and their inclusion in the training dataset. Simulations not included in the training dataset are used as test datasets.

| Process | PDF (Phase Space Bias) | In Training? |
|---|---|---|
| $t\bar{t}$ | CTEQ6L1 | |
| | CTEQ6L1 (biased) | ✓ |
| $Z$+jets | CTEQ6L1 | ✓ |
| | CTEQ6L1 (biased) | ✓ |
| $W$+jets | CTEQ6L1 | |
| | CTEQ6L1 (biased) | ✓ |
| Dijets | CTEQ6L1 | ✓ |
| | CTEQ6L1 (biased) | ✓ |
| Leptoquark | CTEQ6L1 | |

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
