# Peer review of "Towards Universal Unfolding of Detector Effects in High-Energy Physics using Denoising Diffusion Probabilistic Models"

_SciPost Physics, doi:SciPost Phys. Core 8, 064 (2025)_

## Round 1 · Referee Report · Anonymous (Referee 1) · 2025-2-11

Strengths

  1. Novel idea for improving generative unfolding

  2. Well-chosen test cases to present the new method

  3. Well and clearly written text with excellent grammar

Weaknesses

  1. Too much focus on the already-explored concept of generative unfolding, distracting from the real innovation that is presented

  2. Hard to read and questionably placed figures

Report

The authors of this work present a novel addition to the field of generative, machine learning-based unfolding. The new method conditions the generative unfolding model on the moments of some of the to-be-unfolded observables in addition to the observables themself. This effectively gives the unfolding model access to dataset-level information, in addition to the standard event-level information. The authors further demonstrate that this added information results in a more generalizable unfolding model, which can be applied to a range of different physics processes.

I view this as a new step in the use of machine learning in particle physics, and in my view, it represents a significant step in building more generalized unfolding models. As such, I believe the concept and results presented in this work are a good fit for this journal. However, I also feel that the presentation of both method and results can be notably improved, which is why I would recommend minor revisions before publication. A list of recommended changes can be found below.

Requested changes

For the sake of readability, I will try to break this list down into major and minor requested changes.

Major: 1. L123 Section 2 "Methods". This section describes both the concept of including moment-conditioning in the unfolding, and the DDPM/cDDPM. However, it briefly mentions the moment-conditioning and then only goes into more depth about the conditioning after a longer derivation of the cDDPM. In my view, moment-conditioning is a smart and novel approach to generalizing generative unfolding methods, while the cDDPM is notably less new and exciting. Moreover, I fail to see why the moment-conditioning is presented as so strongly tied to the cDDPM, when it should be applicable to any generative unfolding model. Therefore I would request a reordering of section 2, placing greater and earlier emphasis on the moment-conditioning (moving some of the details from 2.3 into 2.1) and adding a paragraph or two on the transferability of the moment-conditioning concept to other generative approaches.

  1. L144 Section "2.2 Denoising Diffusion Probabilistic Models" does a commendable job of presenting the cDDPM, however, I fail to see how the cDDPM fundamentally differs from other diffusion-based conditional unfolding methods, such as the ones mentioned in the introduction. I would request an added discussion the differences of the cDDPM compared to other diffusion-based Unfolders at the end of Section 2.2.

  2. L275: "Part 2: Generalized Unfolder" This subsection describes the process of training the generalized cDDPM in quite a significant amount of detail, but never seems to specify how the conditional moments are derived during evaluation on a real dataset. The context indicates that one would calculate the moments over the whole evaluation set, but the note that moments are calculated on a process-by-process basis is at odds with this, as this would not be possible on a real dataset. An additional segment clarifying how the evaluation is performed in detail is needed here.

  3. L365: "The dedicated unfolder is trained using data pairs (x, y), excluding in the distributional moments. In contrast, the generalized unfolder is trained on multiple simulated physics processes" and L375: "the generalized unfolder learns to model multiple posteriors from the diverse physics processes in its training data, whereas the dedicated unfolder captures only a single posterior represented by its specific training set." It appears to me that there are two main differences between the two approaches. On the one hand, one model is conditioned on the moments, while the other is not, and on the other, one model is trained on a wide range of processes, while the other is trained only on one. These two effects appear non-trivial to disentangle. While a non-conditional model trained on a wide range of priors could, intuitively, be similar to an approach simply trained on the average prior, the more varied training set can still impact the non-conditional model's training behavior. I would request a toy-case test where the conditional and non-conditional models are trained on the exact same data, to better quantify the relative contributions of the two effects.

  4. L441: "In Figure 8, the model’s efficacy is further demonstrated with two tests: (1) reconstructing jet mass from unfolded results, indicating well-preserved correlations among jet vector components, and (2) reconstructing event-level observables from unfolded quantities, achieved by tracking event numbers through object-wise unfolding. The successful reconstruction of jet mass, which is not directly unfolded but derived from". Reconstructing higher-level observables is a physically well-motivated method of checking how well correlations are learned. However,classifier-based tests, especially using classifiers with inputs consisting of both detector-level and event-level information have become the gold standard for testing correlation correctness. As correctly modeling correlations is a major hurdle for high-dimensional unfolding, I would strongly suggest adding such a classifier test to the paper to better quantify the generalized cDDPM's ability to correctly learn generalized correlations.

  5. Figures 3, 5, 6, and to a lesser extent Figures 1, 2, 4, 7, 8: The font on these figures is very small and challenging to read. Especially for Fig. 3, 5, and 6 the label and legend font is around 30% the height of the main text. This means that on a standard 1080p screen, viewing all 3 panels in these plots side by side already results in significant pixelation of the text and subsequent difficulty in reading the text. Two-panel figures are notably better than the three-panel ones but are still challenging to read in printed format. I would request increasing the font size in all figures and restructuring Fig. 3, 5, and 6 to only have two panels, (potentially adding an additional figure)

  6. Both table 1 and the conclusion compare the methods presented to other unfolding methods, however, there is never a quantitative performance comparison between the presented method and other ML unfolding approaches. While I would love to see either other unfolding methods tested in the benchmarks used in the paper, or the generalized cDDPM tested on e.g. the OmniFold dataset, I understand that this is not a reasonable request at this point. In lieu, I would request an added point in the conclusion more explicitly addressing this lack of a quantified comparison, with a bigger focus on the novelty of the moment conditioning concept.

Minor 8. L59 "Related Work" currently is missing a discussion of ML unfolding that already sees application to real experimental data, such as at ATLAS (2405.20041), LHCb (2208.11691), and especially H1 (2108.12376, 2303.13620). Adding this would better show the relevance of the work to current HEP experiments.

  1. L90 "Such a method would enable the unfolding of distributions for a wide range of processes, including those involving yet-undiscovered particles in new physics searches at high-energy colliders." This sentence strikes me as somewhat misleading, as even a perfectly general SM-trained unfolding model still would not be guaranteed to be able to unfold all BSM particles. I would suggest a change to "Such a method would enable the unfolding of distributions for a wide range of processes, including a subset of those involving yet-undiscovered particles in new physics searches at high-energy colliders."

  2. L124. The section title "Our Unfolding Approach" strikes me as too colloquial. Maybe something like "Generalizing Unfolding Approaches" would be more fitting.

  3. L203: "This bias is undesirable for the unfolding task" -> "This bias can prove detrimental for the unfolding task"

  4. L263 "is acceptable or even desirable" -> "is acceptable or even desired".

  5. L314 "While both approaches showed promising results, the latter demonstrated marginally better performance and was therefore adopted for all results presented in this work." The fact that the inclusion of the conditional moments in the output results in a performance increase strikes me as counterintuitive, as it appears to increase the difficulty of the generative task. A sentence or two discussing possible reasons for this behavior would make this a lot clearer.

  6. Figure 8: The left-hand panel seems to have a different format compared to all other figures of this type for no apparent reason. This is no issue if intentional, but I wanted to raise it in case this is a case of an old figure that should have been updated.

  7. General: the positioning of the figures across the paper should be improved. Especially having the conclusion broken up by two pages of results plots breaks the flow of the paper.

Recommendation

Ask for minor revision

  • validity: high
  • significance: good
  • originality: high
  • clarity: high
  • formatting: reasonable
  • grammar: excellent

Author:  Camila Pazos  on 2025-06-09  [id 5551]

(in reply to Report 1 on 2025-02-11)

Thank you for your feedback! All the minor changes have been implemented, as well as all the changes to the plots to make them more visible. With the changes that were made to the text, the figures have re-positioned themselves so that the conclusion is no longer broken up plots, instead all the plots are contained within section 3. Let me know if you think this requires further editing!

For the major changes you suggested, we have implemented most of them and wanted to address the comments, directing you to the changes made or clarifications:

1) L123 Section 2 "Methods" We dedicate Section 2.2 to the specific implementation of the cDDPM because the moment-conditioning approach requires a careful consideration of the underlying generative model. The moment-conditioning approach does not work with any generative approach, including the standard guided conditioning method commonly used in conditional diffusion models. The guided conditioning approach explicitly evaluates the prior distribution, introducing systematic bias toward the mixed training dataset, which would directly counteract our moment-conditioning strategy preventing the model from properly generalizing to unseen distributions based on the provided moments. We consider our choice of direct conditioning (without guidance) to be essential rather than arbitrary, since it allows the distributional moments to serve as the primary conditioning mechanism with minimal interference from training priors. So the technical description of the cDDPM is meant to highlight a key aspect of our contribution that enables the generalization. We have added some clarifying sentences to the end of sections 2.1 and 2.2 to better link the moment-conditioning concept to the specific requirements for the generative model implementation. Please let us know if you believe this requires further edits or clarification!

1) L144 Section "2.2 Denoising Diffusion Probabilistic Models" We have added a paragraph at the end of Section 2.2 that compares our cDDPM approach to other diffusion-based unfolding methods (VLD, DiDi) and other general generative unfolding methods, clarifying the key technical differences that enable our generalization capabilities.

2) L275: "Part 2: Generalized Unfolder" There may be a misunderstanding about how moment-conditioning works on real data. During training, we calculate moments process-by-process only to teach the model how different physics processes are characterized by different moment values. During evaluation on real experimental data, we simply calculate the six moments from the entire dataset we wish to unfold - regardless of what underlying physics processes it contains. The model then uses these moments to determine the appropriate posterior for unfolding. This is one of the advantages of our approach: we don't need to know a priori what physics processes are present in the data. We’ve added some clarifying statements in Section 2.3 to make this evaluation procedure more explicit.

3) L365: Tangled effects of diverse training data and moment conditioning The test you suggest is actually presented in Figure 4. Figure 4(a) shows a generalized unfolder trained on the same diverse training dataset but without moment conditioning, while Figure 4(b) shows the same setup with random "fake" moments. When compared to Figure 3(c), which uses the same training data with proper moment conditioning, the performance difference demonstrates that the moment conditioning itself (rather than simply training on diverse data) is key to enabling generalization. We have added some lines to the last paragraph of section 2.3 to clarify that Figure 4 specifically addresses this disentanglement of effects.

4) L441: Classifier-based testing for correlations We agree that classifier-based tests would provide a much better assessment of the correlations preservation! In this introductory study we focused on exploring the generalization capability (handling unknown processes) and decided to use the reconstruction of derived quantities as an initial validation of correlation preservation. For future work, we will definitely implement a classifier-based study to better understand how this method handles correlations.

6) Quantative comparison to other methods Our team also discussed this point of direct comparisons to other methods throughout the course of this research. In the end, we thought that other methods solve fundamentally different problems (process-specific vs. process-agnostic unfolding), making direct comparisons non-insightful. We have added a paragraph in the Conclusion explicitly addressing this and emphasize that our contribution's novelty lies in the moment-conditioning concept enabling unprecedented generalization capability rather than superior performance on individual known processes, where other models excel.

---

## Round 1 · Referee Report · Anonymous (Referee 2) · 2025-3-14

Strengths

  1. New method to solve the bias problem in unfolding
  2. Variety of different processes are included
  3. For the dataset used in this study, the results of the generalized unfolder look promising

Weaknesses

  1. Dataset lacks exciting correlations
  2. Not clear how one dimensional metrics can quantify performances in a way meaningful to experimental analyses
  3. Lack of comparison to ml-methods that correct for prior dependence
  4. Missing distinction between own contribution and existing work

Report

An existing challenge of classical and ML-based unfolding algorithms is their lack of generalisability. This paper demonstrates some interesting follow-up work to existing machine learning based unfolding algorithms. It provides a method to unfold individual QCD jets in a generalized and unbiased way without retraining.
There are many interesting physics processes that suffer from prior-dependent unfolding results, especially when looking at resonant phase spaces. The chosen dataset also shows diverse pt-distributions for different physics processes, however it lacks exciting correlations. Instead of choosing jets from interesting physical processes, the study was done one QCD jets, a seemingly easy case study.
The authors tried to quantify the improvement of the generalized unfolder by one dimensional metrics, it is unclear how this translates to an improvement in a downstream tasks.
The authors only compare against dedicated generative unfolder without iterative corrections. However, there exists generative as well as discriminative unfolding techniques that iteratively correct for prior dependence. They would provide a more meaningful benchmark.
Lastly, in its current form the paper lacks distinguishability between existing work and own contribution.
Although I think the conceptual idea of the paper can be a good fit for this journal, I would recommend to revise and resubmit.

Requested changes

  1. Introduction: 1.1. The generative unfolding pipeline is already established in many different previous works interchangeably using various generative networks. Using a different generative network itself should not be the selling point for this work. I would therefore ask the authors to stress this when talking about “our contribution”. The crucial novelty of the approach is the inclusion of additional information which produces unbiased results. Also, benefits of training data augmentation has shown to be beneficial in 2105.09923 and it should therefore be cited. 1.2. There has been iterative approach to generative unfolding in [11] to mitigate prior dependencies. Although cited in Table 1, it is listed as a non-iterative method, which is wrong. 1.3. Table 1 also lacks the Omnifold contribution to generalisability 2105.09923 1.4. Throughout the text there was some confusion between object-wise and event-wise unfolding. As far as I understood the authors claim to unfold individual jets rather than an entire event. Is this assumption correct? This generally differs from existing ml-based unfolding studies that unfold event-wise. But then, in Table 1 the authors method is listed as event-wise unfolding. Could this be clarified? 1.5. Although Fig. 1 intends to motivate the success of the method it awkwardly disrupts the flow and the reader does not have enough information yet. I would therefore recommend the authors to keep the results to section 3.

  2. Methods: 2.1. As the authors never actually use unconditional DDPMs, it may be sufficient to stick to conditional DDPMs. 2.2. Again, conditional DDPMs have been used in - and outside of hep. I would therefore ask the authors to cite the respective papers, e.g. 2303.05376, 2307.06836, 2305.10475 etc. 2.3. Why restrict the study to QCD jets only? Please clarify. 2.4. The plot legend is rather small and the ratio plot could be between 0.9 and 1.1 to actually visualize deviations better.

  3. Application to Particle Physics Data: 3.1. As far as I understand the authors over constrain the physical phase space, unfolding (E,px,py,pz,pT, phi, eta) for each jet + the first six moments of the pT-distributions? What happens to the additional degrees of freedom after unfolding? Is pT calculated from the unfolded px and py distribution or directly from the network output? Please clarify in the text. 3.2. The first six moments of the pT-distribution are unfolded as well and thrown away after unfolding. Out of curiosity do they match the truth moments? 3.3. In Figure 8a, it seems that the uncertainty of the dedicated unfolder is within the allowed 3-5% uncertainty region and in a similar region as the generalized unfolder? Do you only observe differences in energy and pt? 3.4. Figure 8b shows an event-wise correlation between two jets. Although, the author’s method is based on individual jets unfolding the unseen correlation is unfolded to a high precision. I would guess this means that detector effects of the dijet system factorize to the two individual jets? Is this true for every correlation? And would this be different for other physical objects? 3.5. I would also suggest the authors to define the hadronic recoil in the text. 3.6. What detector simulation produced the unknown sample of Figure 1a? Please clarify.

B.3 Detector Smearing and Jet Matching B.3.1. Which setting were used to cluster the jets? Maybe clarify in the text.

Recommendation

Ask for major revision

  • validity: -
  • significance: -
  • originality: -
  • clarity: -
  • formatting: -
  • grammar: -

Author:  Camila Pazos  on 2025-06-09  [id 5552]

(in reply to Report 2 on 2025-03-14)

Thank you for the comments and feedback! We have implemented most of the changes suggested and would like to respond to some of the comments where we wanted to add clarification or did not feel the changes were necessary.

1) Introduction 1.1) We have revised the "Our Contribution" section to emphasize that the main novelty is the incorporation of statistical moments as conditioning information. We have kept the mention of the specific generative diffusion model we implement because it is an essential choice. The moment-conditioning would not be successful if implemented with the more common generative networks that use explicit evaluation of the prior during the training. We have also included a reference in “Related Works” to the training data augmentation study you cited. However, we’d like to point out an important distinction: the work presented in 2105.09923 shows a preservation of BSM signals present in data by augmenting the Generation sample with BSM events. Our approach focuses on a different challenge: creating a universal unfolder through moment conditioning that can handle diverse physics processes without prior knowledge of what those processes might be. Our method does also involve training on diverse data, but it is the moment conditioning across the diverse data that enables our generalization capability.

1.3) We have revised the Table 1 caption to define "Generalizable" more precisely. This column is meant to indicate whether a method can train a single model to unfold various physics processes (known or unknown) without retraining or prior knowledge of the underlying physics. The 2105.09923 paper improves phase space coverage by including BSM events in the generation sample, however this constitutes data augmentation rather than generalization to unseen posteriors. In that study, OmniFold still requires process-specific knowledge (knowing which BSM physics to include) and cannot handle truly unknown processes, so we did not feel it falls under the definition of "Generalizable" we use throughout the paper.

1.4) You are correct that our method performs object-wise unfolding of individual jets. We can then achieve event-wise reconstruction by tracking the event number for each unfolded object, as demonstrated in Figure 8b with hadronic recoil reconstruction. We have updated the Table 1 caption to clarify that the "Event-wise" column encompasses both direct event-wise methods and object-wise methods that preserve event-level information through object tracking (our approach falls into the latter category). Otherwise I have double checked that throughout the paper, we consistently refer to our method as "object-wise unfolding," which is the accurate description of our approach.

2) Methods 2.1) We believe the brief introduction to unconditional DDPMs provides some background for readers unfamiliar with diffusion models before introducing the conditional variant we employ. This section is brief and meant to facilitate the discussion of the conditional DDPM and how it differs from the more common guided-conditioning DDPM.

2.2) We appreciate the references to conditional DDPM applications in HEP, but within the scope of this paper we are aiming to focus specifically on unfolding methods rather than the broader landscape of diffusion model applications in physics. The cited papers mostly address event generation, which is a fundamentally different problem from unfolding and we do not wish to dilute the focus on unfolding methodologies that are directly relevant to our contribution.

2.3) We focused on QCD jets to demonstrate our method's capabilities because they are abundant in LHC data and provide sufficient statistics for training. So this seemed like a suitable object with which to begin our studies. Our method requires separate training for different object types (jets, leptons, photons, etc.), but the moment-conditioning framework is directly applicable to any other physics objects. As mentioned in our Conclusion, we do plan to extend the application of this method to other particle types in future work! We have added a line to “Our Contribution” stating that this study focuses on QCD jets, though the method is application to other physics objects as well.

3) Application to Particle Physics Data 3.1) We deliberately overconstrain the kinematic phase space because we found this improves network training performance by providing additional kinematic information. The network outputs all components directly without enforcing kinematic constraints (i.e. pT is not calculated from the unfolded pX and pY, it is directly output by the model). There are small inconsistencies between redundant variables that we viewed as an uncertainty in the unfolding. We have added some lines in section 3.1 clarifying this.

3.2) That's an excellent question that we're also curious about! We did not thoroughly analyze the distribution of the unfolded moments in this study due to time constraints, though at first glance they did appear to be more similar to the truth-level values than the detector-level values. But we plan to look into this!

3.3) Yes, both dedicated and generalized unfolders perform within the typical 3-5% unfolding uncertainty range. The advantage of the generalized unfolder is not necessarily superior performance for known processes compared to the dedicated unfolder, but rather its ability to handle unknown processes without retraining (as we show in Figure 1). As we state in the text, our goal here was to show that our generalized approach doesn't sacrifice accuracy while gaining flexibility.

3.4) This is an excellent point, thank you for bringing it up! Yes we did also assume that the detector effects factorized to the individual jets in this case, however it is true that this might not hold for other correlations or physical objects. We would like to look into this more in future work, and I’ve added lines to our conclusion stating this.

We would like to address some points from the reviewer’s overall report:

Regarding generalization vs. bias correction: The iterative methods you mention that correct for prior dependence address a different problem than ours. Iterative bias correction still requires knowing what physics processes to expect and training separate models. Our contribution is process-agnostic generalization, i.e. a single model that can unfold completely unknown physics without any prior knowledge or retraining. To our knowledge, there are no other unfolding methods that can unfold a process without any knowledge of its prior included in the training (as we demonstrate in Figure 1 with graviton production), so there is no meaningful benchmark to which we can compare these generalization results.

Regarding distinguishability of contribution: We have clarified throughout our revisions that the key innovation is moment conditioning for generalization, not the use of cDDPMs. This enables a level of generalization that existing methods cannot achieve.

---

## Round 2 · List of Changes

Content and Structure Changes - Revised "Our Contribution" section to emphasize moment-conditioning as the main novelty rather than just using a different generative network - Enhanced Section 2.1 and 2.2 with clarifying sentences linking moment-conditioning to specific generative model requirements - Added paragraph at end of Section 2.2 comparing cDDPM approach to other diffusion-based unfolding methods - Clarified evaluation procedure in Section 2.3 for how moments are calculated on real experimental data - Added lines to Section 2.3 clarifying that Figure 4 demonstrates the importance of moment conditioning vs. diverse training data - Added paragraph in Conclusion addressing lack of direct quantitative comparisons to other ML methods and emphasizing moment-conditioning novelty - Added reference to the OmniFold training data augmentation study (2105.09923) in Related Works section - Moved Figure 1 from Introduction to Section 3 to improve flow

Figure Improvements - Increased font sizes in all figures for better readability - Converted three-panel figures to two-panel (Figures 3, 5, 6) for clarity

---

## Editorial Decision

published